# Diverse roles of the metal binding domains and transport mechanism of copper transporting P-type ATPases

Zongxin Guo [1], Fredrik Orädd [2], Viktoria Bågenholm[1], Christina Grønberg[1], Jian Feng Ma [3], Peter Ott [4], Yong Wang [5], Magnus Andersson [2], Per Amstrup Pedersen [6], Kaituo Wang [1,8] ✉ & Pontus Gourdon [1,7] ✉

Copper transporting P-type ($P_{1B-1}$-) ATPases are essential for cellular homeostasis. Nonetheless, the E1-E1P-E2P-E2 states mechanism of $P_{1B-1}$-ATPases remains poorly understood. In particular, the role of the intrinsic metal binding domains (MBDs) is enigmatic. Here, four cryo-EM structures and molecular dynamics simulations of a $P_{1B-1}$-ATPase are combined to reveal that in many eukaryotes the MBD immediately prior to the ATPase core, $MBD^{-1}$, serves a structural role, remodeling the ion-uptake region. In contrast, the MBD prior to $MBD^{-1}$, $MBD^{-2}$, likely assists in copper delivery to the ATPase core. Invariant Tyr, Asn and Ser residues in the transmembrane domain assist in positioning sulfur-providing copper-binding amino acids, allowing for copper uptake, binding and release. As such, our findings unify previously conflicting data on the transport and regulation of $P_{1B-1}$-ATPases. The results are critical for a fundamental understanding of cellular copper homeostasis and for comprehension of the molecular bases of $P_{1B-1}$-disorders and ongoing clinical trials.

Copper is an essential microelement for all organisms and copper-dependent enzymes participate in a palette of physiological processes such as redox reactions, energy metabolism and antioxidative defense[1]. However, strict regulation is necessary as elevated levels of copper cause multiple forms of cell damage, including membrane lipid peroxidation and DNA damage[2]. Copper transporting P-type ATPase proteins are key regulators of cellular copper and are found in all kingdoms of life[3]. The two human members, hATP7A and hATP7B, as well as their prokaryotic CopA counterparts, decrease cytosolic copper levels[4,5]. In eukaryotes, these transporters in addition contribute to metallation of cuproenzymes through cellular transport to the trans-Golgi-network. Physiologically, hATP7A in enterocytes delivers the metal (including that from dietary uptake) to the blood stream for further distribution to other tissues, while hATP7B in hepatocytes excretes copper into the bile for removal from the body[4].

Dysfunctional hATP7A and hATP7B are the direct causes of the severe Menkes and Wilson's diseases, respectively[6,7], underscoring the medical relevance of these proteins. The equivalent proteins in plants, such as rice, are less well studied, yet they have been shown to have roles in copper supply for ethylene signaling and photosynthesis, as well as for sequestration in vacuoles, thereby limiting copper accumulation in the grain[8]. The latter is of potential biotechnological value, for decontamination of copper-polluted soils, or for enrichment of the metal to fulfill dietary needs[9]. Underscoring the significance of such ambitions, rice alone represents a staple food for some 4 billion people globally, providing 21% of global human per capita energy and 15% of per capita protein[10].

P-type ATPases constitute a large group of membrane-embedded, ATP-dependent, active transporters, that build up and maintain concentration gradients across membranes. The copper transporters

[1]Department of Biomedical Sciences, Copenhagen University, Copenhagen, Denmark. [2]Department of Chemistry, Umeå University, Umeå, Sweden. [3]Institute of Plant Science and Resources, Okayama University, Okayama, Japan. [4]Medical Department of Hepatology and Gastroenterology, Aarhus University Hospital, Aarhus, Skejby, Denmark. [5]College of Life Sciences, Zhejiang University, Zhejiang, China. [6]Department of Biology, University of Copenhagen, Copenhagen, Denmark. [7]Department of Experimental Medical Science, Lund University, Lund, Sweden. [8]Present address: State Key Laboratory of Plant Diversity and Specialty Crops, Institute of Botany, Chinese Academy of Sciences, Beijing, China. ✉e-mail: kaituo@sund.ku.dk; pontus@sund.ku.dk

belong to the heavy metal-specific $P_{1B}$-subclass ($P_{1B}$-ATPases), as part of the $P_{1B-1}$-ATPases subfamily, and are further denoted Cu$^+$-ATPases[11]. All P-type ATPases consist of three cytosolic domains designated A-(actuator), P-(phosphorylation) and N-(nucleotide-binding), as well as a transmembrane M-domain with six core transmembrane helices (M1-M6)[3]. Two additional helices, MA and MB, together with an amphiphilic platform, MB', that is present at the intracellular membrane interface in between MB and M1, are specific for $P_{1B}$-ATPases (Fig. 1A)[3]. The P-domain harbors an invariant DKTGT motif, with an aspartic acid that becomes phosphorylated from ATP and then dephosphorylated during the transport cycle (Supplementary Fig. 1A)[12]. The N-domain offers a site for nucleotide binding, preparing ATP for phosphorylation of the P-domain[12]. The A-domain holds an omnipresent TGE motif that is responsible for dephosphorylation[13]. The A- and P-domains are directly linked to the M-domain via M2/M3 and M4/M5 linkers, respectively, while the N-domain represents an extension of the P-domain. Moreover, $P_{1B}$-ATPases harbor so-called metal-binding domains (MBDs), typically positioned in the N-terminus, the number of which vary from zero in certain prokaryotes to six sequential domains in hATP7A and hATP7B[14]. The fact that only one or two MBDs are typically found in prokaryotic members hints that the two domains closest to the ATPase core may be more critical for function, or that distinct molecular principles apply between eukaryotic and prokaryotic proteins[15]. Indeed, MBDs have been proposed to affect cellular localization/trafficking in eukaryotes[16,17]. MBDs commonly have a βαββαβ ferredoxin-like fold, with a solvent-exposed CXXC-motif for metal binding[11,18]. Traditionally, MBDs have been numbered starting at the N-terminal end, meaning that, for N-terminal MBDs, the one closest to the ATPase core has the highest number. However, here we present a new naming convention for MBDs (see further below), where they are numbered from the P-type ATPase core to the periphery, in a sequence direction to make comparisons between proteins with different numbers of MBDs more intuitive. Thus, for N-terminal MBDs, the one closest to the core would be MBD$^{-1}$, the next MBD$^{-2}$ and so on. Conversely, in the cases of C-terminal MBDs these would then have positive numbering, starting at MBD$^{+1}$ closest to the core. This nomenclature will be used throughout.

The transport mechanism of P-type ATPases is described by the Post-Albers cycle[19,20] and is characterized by four cornerstone states, E1, E1P, E2P, and E2, where P marks phosphorylated states (Fig. 1B). Uptake and release of the transported cargo is coupled to phosphorylation and dephosphorylation, as orchestrated by structural rearrangements of the cytosolic A-, N- and P-domains. In the E1 state, the M-domain of the protein is inward-open to allow the cargo to associate, and the N-domain binds ATP. In the E1P conformation, the M-domain closes to an inward-facing occluded state, and the P-domain has become phosphorylated. The shift to the E2P configuration in which the cargo is off-loaded triggers an outward-open M-domain, and displacements of the soluble domains, bringing the A- and P-domains closer together. Next, dephosphorylation generates an outward-facing closed state of the M-domain in E2. Finally, the P-type ATPase returns to the E1 state, and the transport cycle can be re-initiated.

Considering that $P_{1B-1}$-ATPases transport copper in the reduced form, Cu$^+$, Cys/Met residues are preferentially expected to be involved in the process, and the CPC- (in M4), YN- (M5) and MXXXS- (M6) motifs have been shown to serve critical roles for copper transport. These amino acids have been proposed to form one (Cys/Cys/Met-based)[21] or two (Cys/Cys/Tyr and Asn/Met/Ser)[22] occluded high-affinity binding sites in the M-domain, coupled to phosphorylation. Two separate routes have been proposed for the release of copper to the extracellular (or organellar luminal) side where copper chelating proteins may accept the metal ion; one via a narrow pathway near the surrounding membrane[23] or, alternatively, via a larger conduit in the center of the M-domain[24]. Conversely, ion uptake is likely achieved via a transient entry site, involving a conserved methionine (in M1)

adjacent to MB', but different entry-site models have been suggested, including also a nearby glutamate and an aspartate (Supplementary Fig. 1a)[21,25–27]. Delivery to the entry site(s) is also poorly understood. As essentially no copper exists in free form in the cytoplasm, the metal is primarily donated from metallo-chaperone proteins, often with a similar fold as the MBDs, such as ATOX1 in humans, and/or small vehicle molecules such as glutathione[28,29]. It is not fully understood if this process involves the MBDs, highlighting that the functional role(s) of these domains remains enigmatic, with conflicting reports on their role in metal binding, transport and autoinhibition[30–32].

Hitherto, structural information on Cu$^+$-ATPases is limited to the E2P and a subsequent E2.P$_i$ transition state of dephosphorylation of CopA from *Legionella pneumophila* (LpCopA)[23,25], the E2P state of ATP7B from *Xenopus tropicalis* (XtATP7B)[24], and an E1 state of CopA from *Archaeoglobus fulgidus* (AfCopA) and hATP7B[26,33]. These structures generally lack the MBDs, but the structural information for XtATP7B revealed the presence of the two MBDs closest to the core, MBD$^{-2}$ and MBD$^{-1}$ (previously MBD5 and MBD6, respectively), while the hATP7B structure displayed presence of one MBD (see more below). Both the MBDs of the XtATP7B structures were positioned distal to the ion uptake region, with MBD$^{-2}$ bound to the interdomain junction between the A- and P-domains, and MBD$^{-1}$ to the A-domain[24]. Functional data hinted that MBD$^{-2}$ and MBD$^{-1}$ were directly involved in copper transfer to the core, and that the peripheral four MBDs, MBD$^{-6}$-MBD$^{-3}$, were autoinhibitory[24]. Conversely, the MBD of the hATP7B structure was present at the MB' platform. Collectively, while structural data are starting to accumulate, large portions of the $P_{1B-1}$-ATPase transport mechanism remain elusive, limiting our understanding of how ion uptake, transport and release are achieved and regulated, especially for eukaryotic members.

To better understand the molecular determinants that govern $P_{1B-1}$-ATPase activity in cells, we here report structures of a plant Cu$^+$-ATPase, which combined with functional data and in silico analysis, shed critical light on the molecular function of these cellular pumps.

## Results and discussion
### Overall structure

To further dissect the transport and regulation mechanisms of $P_{1B-1}$-ATPases, we focused on a member from rice (*Oryza sativa*), OsHMA4, or simply HMA4[9]. This protein is localized to the tonoplasts of roots, and is involved in root-to-shoot translocation and grain accumulation of copper. HMA4 has three predicted N-terminal ferredoxin-like MBDs, although the one closest in sequence to the M-domain and the P-type ATPase core, MBD$^{-1}$, lacks the metal-binding CXXC-motif. The protein was produced and purified from *Saccharomyces cerevisiae*, frozen on grids in the presence of detergent and with one of the established state stabilizing phosphate mimics BeF$_3^-$ or AlF$_4^-$ that inhibit protein function (Methods and Supplementary Fig. 2a, b). These have previously been used to trap the outward-open E2P conformation, and an outward-facing (E2.P$_i$) transition state of dephosphorylation, respectively[34]. Separately, we instead prepared the sample with a combination of copper and reducing agent, with the ambition to recover an early E1 conformation. We also studied the effect of no such supplementation to HMA4, which has previously yielded an early E1 state of AfCopA[26]. These various conditions yielded four structures, determined at overall resolutions of 3.3 (HMA4$^{BeF}$), 3.2 (HMA4$^{AlF}$), 3.6 (HMA4$^{apo}$), and 3.7 Å (HMA4$^{Cu}$) (Fig. 1C, Supplementary Figs. 3–7 and Supplementary Table 1). The M-domain was well-resolved in all structures, but the resolution of the soluble domains varied (e.g. the A-domain ranging from well-defined in HMA4$^{BeF}$/HMA4$^{AlF}$, to only assigning the overall position in HMA4$^{apo}$/HMA4$^{Cu}$), and we modeled up to residues 184–957 out of 978 amino acids (Supplementary Figs. 8–12). As anticipated, the $P_{1B}$-ATPase architecture was maintained, with three soluble core A-, P- and N-domains, and a transmembrane M-domain composed of eight transmembrane helices (MA,

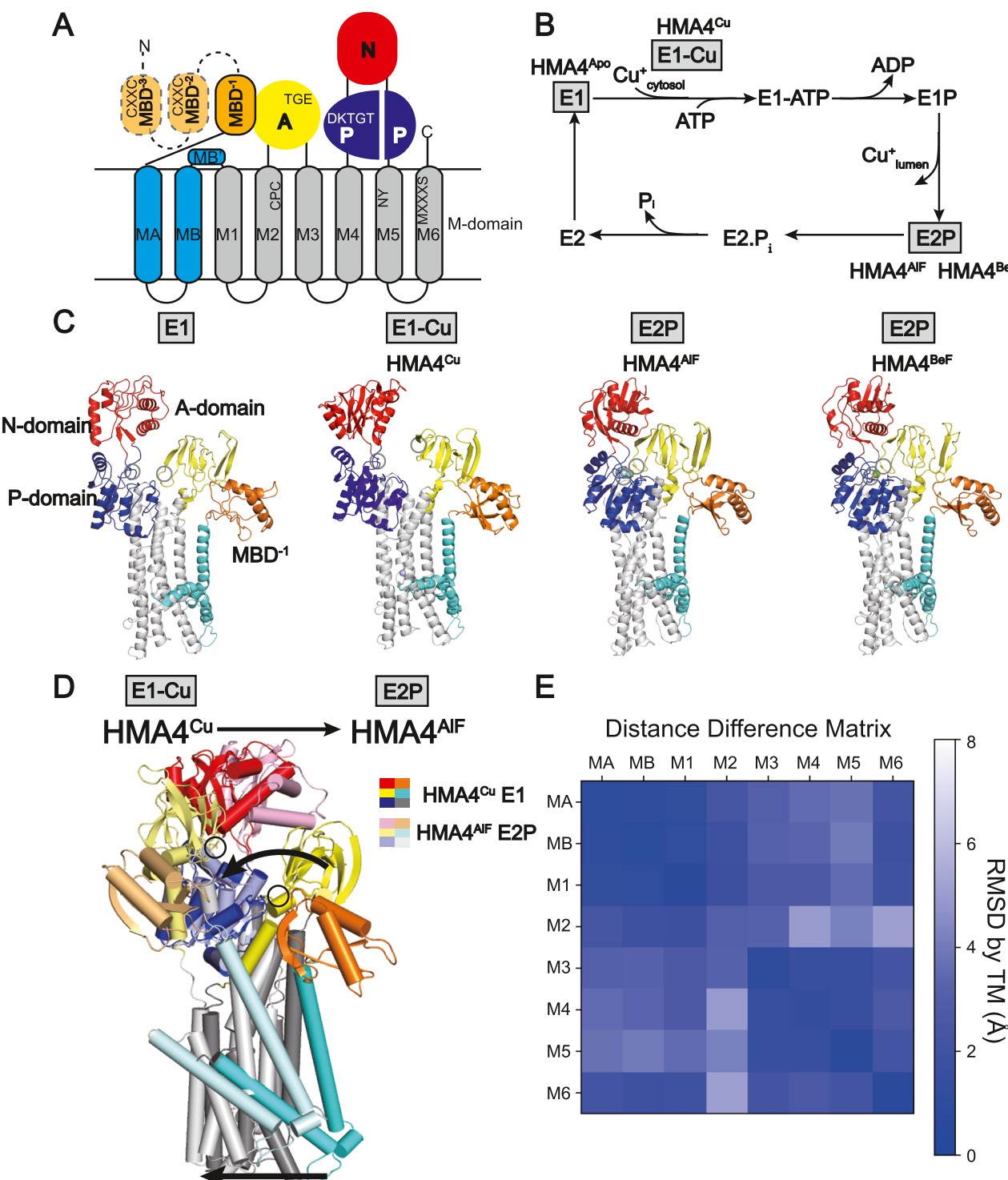

**Fig. 1 | Topology and E1-E2P-E2P-E2 Post-Albers transport cycle of copper transporting P-type ATPases, as well as structures of the HMA4 member from rice. A** Topology with the A- (actuator, yellow), P- (phosphorylation, blue), N- (nucleotide binding, red) and M-domains (transmembrane domain, cyan and gray), and 0–6 metal binding domains (MBDs, orange, 3 in HMA4). **B** Schematic transport mechanism with inward- (E1) and outward-facing (E2) states that undergo auto(de) phosphorylation to E1P and from E2P, respectively. The structurally determined conformations are highlighted in gray. **C** Cartoon representation of the cryo-EM structures of the HMA4 intermediates determined in this work, colored as in panel

A. The TGE and DKTGT-motifs are shown with black circles. **D** Structural changes associated with the shift from the E1-Cu (HMA4$^{Cu}$) to the E2P (HMA4$^{AlF}$) state, colored as in panel A and in lighter shades, respectively. The circles represent the TGE-motif sites of the A-domain. Arrows highlight the changes. **E** Distance difference matrix in-between the E1 (HMA$^{apo}$) and E2P (HMA$^{AlF}$) states, showing the relative movement of each TM helix in the M-domain. The matrix was calculated as in [84]. MA-M1 and M3-M5 form separate rigid bodies. M2 moves considerably compared to M3-M6.

MB, and M1–M6), as well as the MB' platform (Fig. 1C). A striking feature of the structures is that we consistently detect MBD$^{-1}$ (covering residues 184–258), which is directly linked to helix MA via a partially ordered linker, but with no indications of the other two MBDs (see later).

## Inward-open (E1) and outward-open (E2P) structures

Next, we set out to identify the location of these structures in the conformational landscape of P$_{1B}$-ATPases. Global structural alignments suggest the structures are pairwise related (RMSD HMA4$^{apo}$ to HMA4$^{Cu}$ 2.6 Å, and HMA4$^{BeF}$ to HMA4$^{AlF}$ 0.7 Å), with much lower similarity between each pair (RMSDs ranging from 7 to 9 Å, Supplementary Table 2). Notably, the available HMA4 and hATP7B AlphaFold models are more similar to our apo/Cu-pair structures (RMSD approximately 3.4 and 3.6 Å, respectively) to both HMA4$^{apo}$/HMA4$^{Cu}$). Inspection of the cryo-EM HMA4$^{BeF}$ and HMA4$^{AlF}$ maps and models suggest BeF$_3^-$ or AlF$_4^-$ associated with the TGE- and DKTGT-motifs, starting with T480 and D641, respectively (Supplementary Fig. 12). The only noticeable difference between HMA4$^{BeF}$ and HMA4$^{AlF}$ is a subtle shift of the N-domain, while more significant discrepancies are observed between HMA4$^{apo}$ and HMA4$^{Cu}$, mainly in the N- and A-domains, the MBD and MB' (Supplementary Fig. 13). Unexpectedly, analyses suggest that the four structures are quite similar in the proposed ion-uptake region at the cytosolic membrane interface adjacent to MB', M1 and M4 (RMSDs from 0.6 to 3.3 Å for the M-domains, Supplementary Table 2). However, in HMA4$^{BeF}$/HMA4$^{AlF}$ the M3 transmembrane segment is partly covering the MB' platform, and the two cysteines of the invariant CPC-motif of M4 are further away from the intracellular surface than in HMA4$^{apo}$/HMA4$^{Cu}$, indicative of an architecture less compatible with receiving metal (Supplementary Fig. 13). On the other side of the membrane, an outside-exposed pathway is present in HMA4$^{BeF}$ and HMA4$^{AlF}$, but not in the other two structures (Supplementary Fig. 14). In comparison to available structural information on P$_{1B-1}$-ATPases, HMA4$^{apo/Cu}$ are most alike the early inward-open E1 structure of AfCopA (RMSD 4.1 and 3.8 Å)[26], while HMA4$^{AlF}$/HMA4$^{BeF}$ are most reminiscent of the outward-facing structure of LpCopA captured using AlF$_4^-$ (RMSD 1.8 and 1.7 Å, Supplementary Table 2)[24]. Equivalently, the position of the omnipresent TGE- and DKTGT-motifs in the soluble domains shift from distant in HMA4$^{apo}$/HMA4$^{Cu}$ to adjacent to each other in HMA4$^{BeF}$/HMA4$^{AlF}$, the latter representing a requirement to permit dephosphorylation (Fig. 1C). Collectively, these observations suggest the HMA4$^{BeF}$/HMA4$^{AlF}$ structures represent early E2 conformations linked to copper release, likely E2P, while the HMA4$^{apo}$/HMA4$^{Cu}$ rather represent early E1 states associated with copper uptake.

## Conserved E1 → E2P changes among P$_{1B}$-ATPases

It has previously been proposed that ATP turn-over of the soluble domains in P$_{1B}$-ATPases is different from other P-type ATPase classes, but the validity of these studies could be questioned considering the spread in origin of the compared members and because different experimental techniques have been used to obtain the structures[23–26,33]. Our HMA4 structures clearly support that the E1, but not E2P, conformation of P$_{1B-1}$-ATPases is unique. Similar to what is seen for the E1 state of AfCopA[26] the A-domain, and in particular its TGE-motif, adopts a peculiar arrangement in the E1 state compared to the more well-characterized P$_2$-ATPases, including the calcium-transporting member SERCA[35] (Fig. 1D and Supplementary Fig. 13, 15). We note that this difference is independent of the MBDs, as the AfCopA structure was lacking such domains and still preserved the unique A-domain organization. Collectively, this substantiates the argument that the distinct A-domain position rather depends on a linker in-between M1 and the A-domain that is missing in P$_{1B}$-ATPases[26] (Supplementary Fig. 15). The previously detected large conformational changes associated with the E1 → E2P shift as deduced by comparisons of AfCopA and LpCopA[26] are also preserved

in HMA4. The A-domain rotates 100° clockwise relative to the P-domain as seen from the cytosol to enable subsequent dephosphorylation, with a less significant displacement of the N-domain following release of the nucleotide (Fig. 1D)[26]. Analogously, unbiased analyses of the relative locations of each transmembrane helix in the two states revealed that conformational changes of the M-domain occur as two separate helix bundles: MA-M2 and M3-M6[26]. In HMA4, a similar two-helix bundle arrangement is observed, but the helices involved are instead MA/MB/M1/M6 and M3/M4/M5, with M2 located in between these subdomains and moving independently (Fig. 1E). Overall, this speaks for conserved conformational changes during the transport cycle for P$_{1B}$-type ATPases, with the HMA4 structures confirming the particular domain rearrangement of the E1 state being unique to this subfamily, and the E2-states being more similar to other P-type ATPases.

## MBD$^{-1}$ serves as an A-domain extension

A question of outstanding significance is the role of the MBDs in P$_{1B}$-ATPases. As previously indicated, our cryo-EM data displays density for MBD$^{-1}$, but not MBD$^{-2}$ and MBD$^{-3}$, across all four determined structures. Interestingly, the location of MBD$^{-1}$ is maintained relative to the A-domain across transport states. Its position corresponds to the equivalent MBD$^{-1}$ in XtATP7B, structurally determined in the E2P state (Fig. 2A, B)[24], and it is also similar to that of MBD$^{-1}$ in AlphaFold models of many eukaryotic but not prokaryotic P$_{1B}$-members (Fig. 2C and Supplementary Fig. 16). Indeed, inspection of the available cryo-EM maps suggest it also occupies this position in previously determined hATP7B E1 structures, but relatively low resolution can explain why a feature at the MB' platform was instead interpreted as MBD$^{-1}$ (see further below, Supplementary Fig. 17)[33]. Thus, our and previous data together indicate a previously undiscovered location of MBD$^{-1}$ in association with the A-domain that is preserved throughout the transport cycle, and likely broadly conserved across eukaryotic P$_{1B}$-ATPases.

The HMA4 structures reveal that the MBD$^{-1}$/A-domain interaction is based on two conserved salt-bridges, D233-R449 and R238-D451. These pairs are preserved also in XtATP7B, in which the equivalent of R238-D451 (R613-D826 in XtATP7B) has been shown to be important for function (Supplementary Fig. 1b)[24]. Interestingly, while the N-terminus harboring the six metal binding domains constitutes almost half of the protein in hATP7B, only six of more than 100 curated amino acids positions affected by disease-causing missense mutations are located in parts preceding the MA-MBD$^{-1}$ linker[15]. One of these six locations concerns the equivalent of R238-D451 (R616-D829 in hATP7B), suggesting the salt-bridge as well as the MBD$^{-1}$ and A-domain interaction are physiologically relevant also in humans. Indeed, the R616W hATP7B form has even been proposed to completely abolish transport activity[17]. The implication of the peripheral position adjacent to the A-domain is that MBD$^{-1}$ is likely neither directly linked to auto-inhibition, as its position would not hinder turn-over of the soluble domains, nor to copper delivery to the ATPase core, as it is located far away from the MB' platform in all the recovered HMA4 structures. Instead, MBD$^{-1}$ probably serves a structural role, positioning the preceding MBDs, and adding a connection between the core of the A-domain and the M-domain in P$_{1B-1}$-ATPases. As such, MBD$^{-1}$ would rather serve as an A-domain build-on as observed for other N-terminal features in all other major P-type ATPase classes, thus representing a hotspot for tuning P-type ATPase function[36–41]. That MBD$^{-1}$ serves a structural role is further supported by MBD$^{-1}$ of HMA4 lacking the CXXC-motif, despite maintaining the ferredoxin-like fold. In addition, we note that many species across evolution such as other plants, nematodes and fungi have a similar setup as HMA4, with their MBD$^{-1}$ lacking a CXXC-motif (e.g. *Arabidopsis thaliana* UNIPROT-ID B5AXL4, *Caenorhabditis elegans* O17537 and *Neurospora crassa* Q7SGS2). Hence, MBD$^{-1}$ is unlikely to serve an important role due to interaction

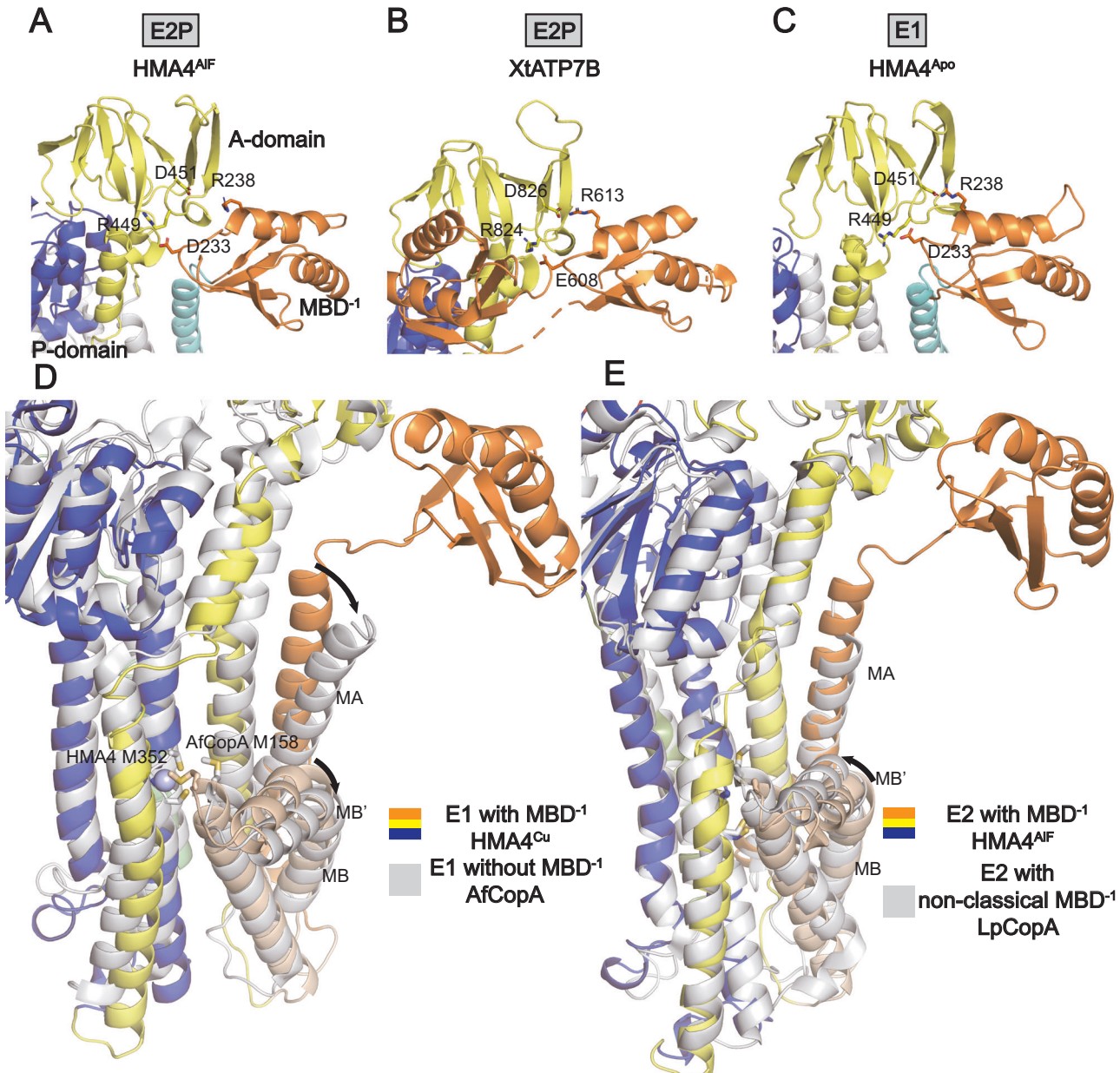

**Fig. 2 | The metal binding domain next to the ATPase core (MBD$^{-1}$) interacts with the A-domain and reshapes the MB' platform. A–C** The local arrangement of the A-domain and MBD$^{-1}$ is maintained across states and species. **A** Close-view of the A-domain-MBD$^{-1}$ interaction in the E2P state (HMA4$^{AlF}$). **B** Close-view of the A-domain-MBD$^{-1}$ interaction in the E2P state (XtATP7B from *Xenopus tropicalis*). **C** Close-view of the A-domain-MBD$^{-1}$ interaction in the E1 state (HMA4$^{apo}$). **D, E** MBD$^{-1}$ affects the MB' platform, with an additive effect from E1 → E2P. **D** Comparison of the E1 states of HMA4 (with MBD$^{-1}$) and AfCopA (structure determined in the absence of MBD$^{-1}$), respectively. Arrows indicate a shift related to absence of MBD$^{-1}$. **E** Comparison of the E2P states of HMA4 (with MBD$^{-1}$) and LpCopA (with a non-classical MBD$^{-1}$), respectively. Arrows indicate a shift related to absence of classical MBD$^{-1}$.

with copper, also in species beyond rice, although copper-binding is permitted in e.g. hATP7B.

## MBD$^{-1}$ remodels the ion uptake region

The above-mentioned M1/A-domain linker is of particular interest as it is critical for function in P$_2$-ATPases, where shorter or elongated connections abolish ion transport, presumably affecting ion uptake and/or turn-over[42,43]. Similar to the situation in P$_2$-ATPases, the length of the linker from the ATPase core to MBD$^{-1}$ is typically rather short[15]. Furthermore, three disease-causing ATP7B mutations are found in the linker between MBD$^{-1}$-MA, congruent with the notion that the linker between the soluble and transmembrane domains is also important in P$_{1B}$-ATPases[15]. Structural comparisons of HMA4$^{apo}$/HMA4$^{Cu}$ with the

equivalent E1 structures of AfCopA, which were determined using protein that lacks N-terminal MBDs[26], suggest a tighter arrangement of MA-M1 relative to M2-M6, and a shorter distance in-between the proposed methionine-dependent entry site (M352 in HMA4) and the MB' platform in HMA4$^{apo}$/HMA4$^{Cu}$. This structural difference is further enhanced in comparison to the equivalent E2P states, assessing HMA4$^{BeF}$ versus the same state of LpCopA that harbors a non-ferredoxin-like MBD[23] (Fig. 2D, E). Thus, the MBD$^{-1}$/MA linker reshapes the metal uptake region in an E1-E2P-dependent manner, presumably priming the structure for copper uptake. This may explain why MBD$^{-1}$ appears critical for function in hATP7A and hATP7B, to enable the appropriate MB' platform design for copper delivery in vivo[15,25].

## Functional analysis of the metal-binding domains suggests multiple functional roles

Next, with the intention to validate our structural studies we strived to functionally assess HMA4 in vitro. However, these efforts were fruitless, despite attempts to characterize the function in different lipid/detergent environments as well as in liposome, which is consistent with our previous experience with certain $P_{1B-1}$-ATPases. The finding that the role of MBD$^{-1}$ is maintained across many eukaryotes demonstrates that metal handling of $P_{1B}$-ATPases is conserved also well beyond the transport across the M-domain, supporting cross-species studies and suggesting that data collected for complementary $P_{1B}$-targets are broadly relevant. The short linker in-between MBD$^{-1}$ and MBD$^{-2}$ (6 residues in HMA4 and hATP7B) further reinforces this argument[15]. Thus, to complement the structures of HMA4, XtATP7B and hATP7B and to build on previous functional analysis of MBDs in e.g. XtATP7B, we functionally characterized hATP7B in vitro, by measuring the production of inorganic phosphate in the presence of copper to assess ATPase activity[26,44]. Akin to HMA4, full-length hATP7B was recovered from *S. cerevisiae* and purified to near homogeneity (Supplementary Fig. 3c). The sample demonstrated clear copper-dependent activity that was sensitive to the inhibitor AlF$_4^-$, indicating that the retained protein is functional (Fig. 3A). Considering the reported stimulatory effect of truncation of MBD$^{-6}$-MBD$^{-3}$ in XtATP7B, and hence an apparent autoinhibitory role of the MBD$^{-6}$-MBD$^{-3}$ cluster[24], we rather focused on comparing full-length hATP7B with two complementary forms to extend beyond the available functional data. We prepared one construct lacking all MBDs Δ(MBD$^{-6}$-MBD$^{-1}$)hATP7B, and one without all but one MBD Δ(MBD$^{-6}$-MBD$^{-2}$)hATP7B, and tested the activity under conditions of abundant copper, when no autoinhibition is expected. Notably, Δ(MBD$^{-6}$-MBD$^{-1}$)hATP7B almost completely abolished the function, while Δ(MBD$^{-6}$-MBD$^{-2}$)hATP7B retained approximately 50% activity. These data hint at a special, crucial, role of MBD$^{-1}$ in eukaryotic $P_{1B-1}$-ATPases, congruent with the structural function suggested from our structures. Conversely, our functional data suggest the mechanistic role of MBD$^{-2}$ is likely non-essential and stimulatory, as removal reduces the function to approximately 50 % of that of wild type, thereby separating MBD$^{-2}$ from MBD$^{-1}$ and the MBD$^{-6}$-MBD$^{-3}$ cluster.

We also exploited an assay based on the ability of $P_{1B-1}$-ATPases to complement the high-affinity iron-uptake deficiency of yeasts lacking the homologous protein Ccc2p[45]. One benefit of the setup is that the experiments are conducted in a native environment, but that renders the data somewhat more difficult to interpret, considering that multiple factors can affect cell growth. Nonetheless, our yeast studies of HMA4 essentially reproduced the consequences of the ATP7B deletions (Supplementary Fig. 18). While wild-type HMA4 and a truncation of MBD$^{-3}$, Δ(MBD$^{-3}$)HMA4, complemented the yeast strain, removal of the first two, Δ(MBD$^{-3}$-MBD$^{-2}$)HMA4, or all MBDs, Δ(MBD$^{-3}$-MBD$^{-1}$)HMA4, did not. This is again indicative of a more significant role of MBD$^{-2}$ and MBD$^{-1}$, and perhaps even an essential role also for MBD$^{-2}$. On the other hand, SXXS-substitution of the CXXC-motif of MBD$^{-2}$ sustained the cell growth, hinting at a more complex role for MBD$^{-2}$. Collectively, these data point towards diverse roles of the MBDs, which could explain the conflicting reports in the literature, and a possibility during evolution to add capabilities using multiple sequential MBDs[46–52]. It is at the same time congruent with the two most ATPase core proximal MBDs (MBD$^{-2}$ and in particular MBD$^{-1}$) being most important for function[15].

## Complementary MD simulations of the MBD domains

To shed further light on the structure and function of the MBDs beyond MBD$^{-1}$, we turned to MD simulations, which have been exploited previously for studies of the positions of MBDs[53]. Considering the shortage of structural information of MBD$^{-2}$, we conducted a series of simulations of HMA4 incorporated into an in-silico

membrane, assessing the E1 (using HMA4$^{apo}$) and E2P (HMA4$^{AlF}$) states obtained using cryo-EM, and different starting positions of MBD$^{-2}$. Throughout, MBD$^{-1}$ was placed in association with the A-domain, as deduced from our structures and XtATP7B[24], and MBD$^{-3}$ was left detached, without interactions with the ATPase core as is the case in the AlphaFold models of HMA4 and hATP7B (Supplementary Fig. 16). Conversely, MBD$^{-2}$ was positioned 1) in association with the MB' platform and the CXXC-motif, essentially facing the entry site methionine, as identified in the HMA4 and hATP7B AlphaFold (E1) models (in the E1 and some of the E2P MD simulation runs), or 2) as in XtATP7B, in-between the A- and P-domains with the CXXC-motif towards the ATPase core (in the remaining E2P simulations). The MBD$^{-2}$ interaction with the MB' platform was generally maintained throughout the E1 and E2P simulations (Fig. 3D–F). However, the position at the MB' platform was more stable in the ten independent E1 simulations we conducted, with final RMSDs between 1.5 and 4.5 Å (mean over the entire simulation time: 3.0 ± 0.82 Å) compared to the starting structure, while in the E2P conformation, the corresponding final RMSDs were 6–9.5 Å (total mean: 5.9 ± 1.0 Å) in five separate runs. Correspondingly, the E2P simulations with MBD$^{-2}$ at the A-/P-domain interface, retained the MBD$^{-2}$ position in three of five runs (final RMSD between 2 and 6.5 Å, total mean: 3.5 ± 0.87 Å), while the domain separated from the starting position in the remaining two simulations. Moreover, while MBD$^{-3}$ started peripheral to the core domains in all MD simulations, in two of the E1 runs the domain interconnected stably with the A-/P-domains for a significant part of the trajectories (Fig. 3D). In all other runs the MBD$^{-3}$ essentially remained unbound, with few interaction points to the ATPase core.

## Proposed functions for the remaining MBD domains

**The position of MBD$^{-2}$ is E1/E2P-dependent, docking to the MB' platform in the E1 state.** How then does the position of MBD$^{-1}$ affect the adjacent MBD$^{-2}$? The E2P XtATP7B structures suggested MBD$^{-2}$ binds in-between the A-/P-domains, partially burying the CXXC-motif towards the ATPase core and thereby likely preventing access from copper-donating chaperones such as ATOX1[24]. This would rather represent a possible model for autoinhibition, in contradiction to our functional results on MBD truncations. However, the domain was significantly less well-resolved than most of the ATPase core in the XtATP7B structures, which may indicate that the proposed interaction is weak. Moreover, mutagenesis of the interface in-between MBD$^{-2}$ and the ATPase left the function essentially unaffected. This may suggest that the identified binding is coincidental, which would be in agreement with our functional data on hATP7B and HMA4 that suggest a stimulatory effect of MBD$^{-2}$. Another argument against an inhibitory effect of MBD$^{-2}$ in the E2P state is that the E2P → E1 shift is associated with dissociation of the A- and P-domains, with the A-domain relocating away from the P-domain, and hence the proposed position MBD$^{-2}$ in the E2P configuration cannot be expected to block the anticipated structural changes (Figs. 1D, 3B).

Instead, based on the MD simulations of HMA4, MBD$^{-2}$ appears to associate with the MB' platform in the E1 states, placing the CXXC-motif towards a methionine which is part of all suggested entry site models for uptake to the ATPase core (M352 in HMA4). Notably, a similar arrangement is present in many AlphaFold models, e.g. of HMA4 and hATP7B (Supplementary Fig. 16). It is also consistent with the E1 state data of hATP7B, which displayed non-core ATPase cryo-EM density at the MB' platform, also in the absence of ATOX1 (PDB-ID 8IOY[33]), although this aspect was interpreted as MBD$^{-1}$ (Supplementary Fig. 17)[33]. As we have already determined that MBD$^{-1}$ is firmly anchored to the A-domain, the well-defined ferredoxin shape located at the MB' platform in the chaperone-free E1 structure of hATP7B is here reinterpreted as MBD$^{-2}$, providing experimental support that MBD$^{-2}$ can dock to the ion uptake region in the E1 conformation. Thus, the position of MBD$^{-2}$ is transport cycle-dependent, located at the MB' in the E1

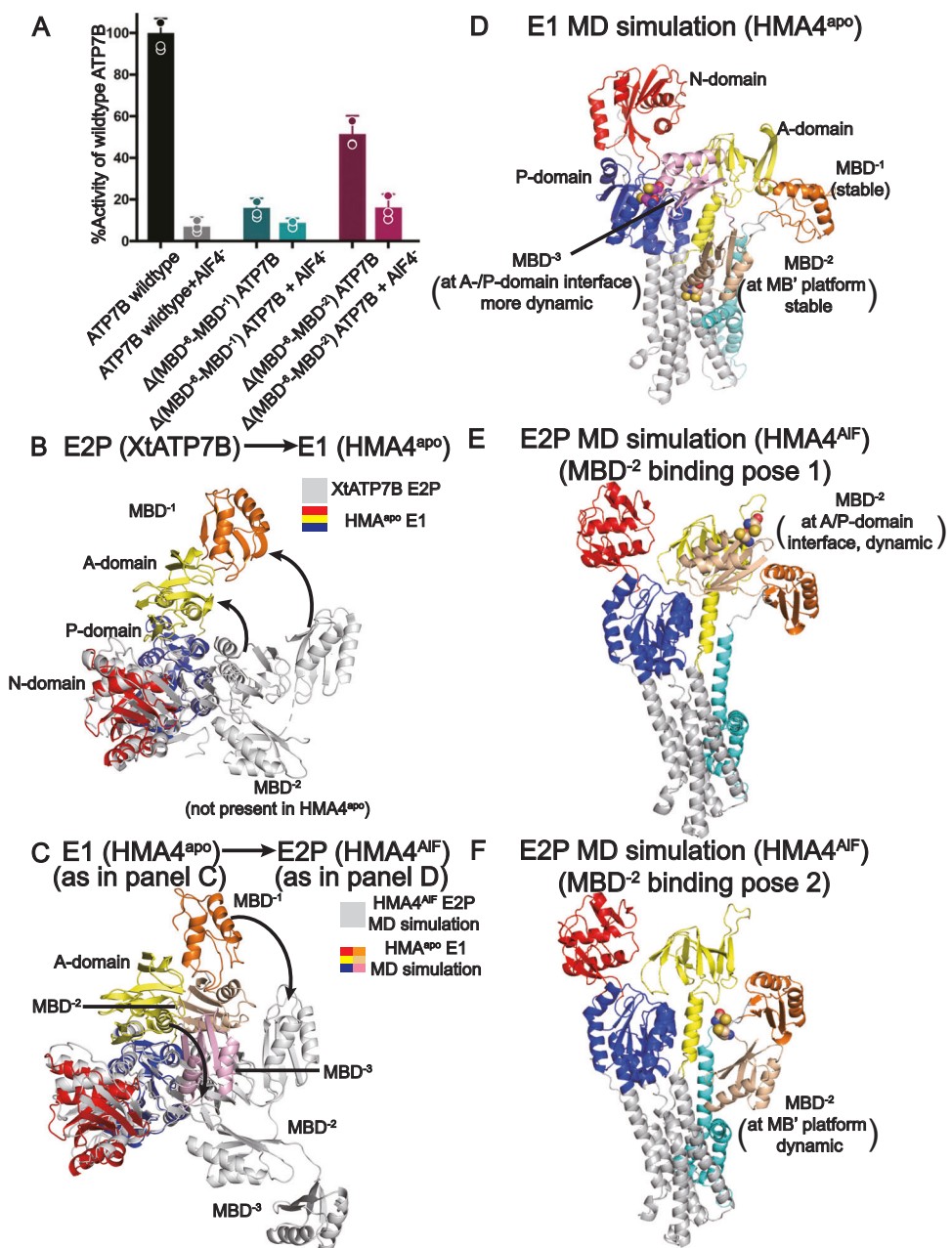

**Fig. 3 | The diverse functional roles of the metal binding domains (MBDs).**
**A** Functional data comparing wildtype human ATP7B and truncations with no MBD
(Δ(MBD$^{-6}$-MBD$^{-1}$)ATP7B) or only MBD$^{-1}$ (Δ(MBD$^{-6}$-MBD$^{-1}$)ATP7B) attached to the
ATPase core. Assessed through detection of released inorganic phosphate in
detergent solution as triggered by copper and ATP in triplicate ($n = 3$ distinct bio-
chemical samples). Data are presented as averages with standard deviation error
bars. See also Supplementary Fig. 2. **B** Structural comparison of the cytoplasmic
domains of the E2P (XtATP7B) and E1 states (HMA4$^{apo}$) suggests the E2P → E1 tran-
sition can occur without MBD$^{-2}$ interference. View from the cytoplasm. **C** Structural
comparison of the E1 (as shown in panel D) and E2P states (as shown in panel E)
suggests the E1 → E2P transition cannot occur without interference of MBD$^{-3}$ when
located at the suggested position. View from the cytoplasm. **D** MD simulations of
the E1 state (HMBA$^{apo}$) indicates MBD$^{-2}$ is stable at the MB' platform, while the MBD
preceding MBD$^{-2}$, MBD$^{-3}$, is flexible, but can interact with the ATPase core at the A-/
P-domain interface. Cysteines of the CXXC motifs shown as spheres. See also
Supplementary Fig. 19. **E, F** MD simulations of the E2P state (HMBA$^{AIF}$) imply MBD$^{-2}$
is dynamic and can interact with the ATPase core at the A-/P-domain interface
(panel D) and at the MB' platform (panel E). Cysteines of the CXXC motifs shown as
spheres in both panels. See also Supplementary Fig. 19.

and at the A-/P-domain interface in the E2P state, a phenomenon that
must be orchestrated by the relatively short linker between MBD$^{-1}$ and
MBD$^{-2}$ and the anchored position of MBD$^{-1}$ to the A-domain.

**MBD$^{-2}$ may assist in copper delivery to the ATPase core.** How can the
stimulatory effect of MBD$^{-2}$ revealed by our truncations be explained?
The arrangement of MBD$^{-2}$ at MB' in the E1 state can be interpreted as a
functional role in metal transfer of cytoplasmic copper to the ATPase

core or autoinhibition. The lack of growth when MBD$^{-2}$ is removed
makes it tempting to speculate that it assists in copper delivery to the
ATPase core. However, the complementing effect of the SXXS-
substitution of the CXXC motif of the HMA4 MBD$^{-2}$ is not consistent
with a direct role in copper delivery, as copper cannot be bound to the
mutant form. This opens for an alternative interpretation of the MBD$^{-2}$
data. Considering the close homology between MBDs and soluble
metallo-chaperones, such as ATOX1, it is likely that the latter also can

associate with the MB' platform for metal delivery, as has been proposed previously[25]. Moreover, it is well established that MBDs and chaperones can interact[54]. Consequently, MBD$^{-2}$ may have a dual role in copper delivery to the methionine at the MB' platform, either directly or indirectly, through the guidance of ATOX1. We note that, in a previously determined MBD-ATOX1 assembly the CXXC-motifs face each other, but the interaction is likely not CXXC-dependent considering the relatively large interaction interface[54] (Supplementary Fig. 20a). This hypothesis is further supported by the disease-causing hATP7B MBD$^{-2}$ mutations L549P[55], that faces the ATOX1 interaction interface, and L492S[56], which reduces ATOX1 interaction (Supplementary Fig. 20a). This notion would also be congruent with all the E1 structures of hATP7B displaying a ferredoxin fold at the MB' platform in the presence of (co-purified) metal, suggesting such conditions stimulate MBD$^{-2}$ or ATOX1 to approach the MB' platform[33]. In line with this, the available experimental data indicate absence of the feature at the MB' platform when copper is not available, such as in our HMA4$^{apo}$/HMA4$^{BeF}$/HMA4$^{AlF}$, all XtATP7B and the apo hATP7B (PDB-ID 7XUN[33]) structures. Why is then MBD$^{-2}$ not visible in our HMA4$^{Cu}$ structure? As we will see later, the hATP7B structures were generated using a mutant form that cannot accept copper, meaning the copper would still be bound to MBD$^{-2}$, while in HMA4$^{Cu}$ the metal has been transferred to the M-domain. This indicates MBD$^{-2}$ remains associated with the MB' platform until copper has been delivered.

**MBD$^{-3}$ may be autoinhibitory.** What about MBD$^{-3}$, which was neither identified in the E2P XtATP7B nor in our own HMA4 structures? The interaction between MBD$^{-3}$ and the A-/P-domain interface seen in two of the E1 state MD simulation runs is reminiscent of, yet distinct from, that observed for MBD$^{-2}$ in XtATP7B in the E2P configuration[24], as the CXXC-motif of MBD$^{-3}$ is somewhat more exposed (Fig. 3B, E). The observation that MBD$^{-3}$ interacts with the A/P-domains in E1 similarly as observed for MBD$^{-2}$ in the E2 structures is peculiar, and in our view validates the results from the MD simulations and highlight a hotspot for MBD interaction with the core of the ATPase. However, in contrast to the position of MBD$^{-2}$ in the E2P configuration, the location of MBD$^{-3}$ in the E1 state will obstruct turn-over of the soluble domains, as the A-domain cannot adopt its new location with the E1 → E2P shift (Fig. 3C). This may at least partially explain the observation made for XtATP7B that the peripheral four MBDs are autoinhibitory and why copper loading of MBD$^{-3}$ in hATP7B has been shown to release autoinhibition[24,32]. It is likely that such regulation occurs when copper levels are low, as turnover would not be required under such conditions, thereby reducing futile ATP consumption. Moreover, autoinhibition may be released through copper binding to the CXXC motif. The fact that MBD$^{-3}$ or any of the preceding MBDs have never been observed structurally in an ATPase core context can relate to low levels of the metal being present in the samples. This could be sufficient to dissociate (MBD$^{-6}$-)MBD$^{-3}$ from the ATPase core, as, normally, not a single free copper ion exists in living cells[57]. The increasing number of domains may collectively add to the copper-sensitivity of such regulation, or assist in other cellular functions such as cellular localization/trafficking. An autoinhibitory role does not exclude a role of the first four MBDs in metal transfer to MBD$^{-2}$. However, the somewhat small functional effect of deletion of the first four MBDs rather favors chaperones such as ATOX1 as copper mediators to MBD$^{-2}$ or to the ATPase core. Indeed, structural analyses of experimental ATOX1-MBD complexes (e.g. PDB-ID 2K1R and 3CJK[54]) indicate ATOX1 is able to provide copper to MBD$^{-3}$ in the position revealed by the MD simulations, thereby presumably also releasing autoinhibition (Supplementary Fig. 20b). Consequently, it is possible MBD$^{-3}$ associates with the A-/P-domain interface in the E1 state, inhibiting transport, and that this regulation is released upon copper binding.

## Copper entry, binding and an alternative role of the invariant motifs in the M-domain

How cargo is accepted from the cytoplasmic surroundings, bound to a transient entry site at the MB' platform, then transferred to one or two high-affinity binding site(s) and finally released to the extracellular side in P$_{1B-1}$-type ATPases remains debated. Before this work, E1 structures of a P$_{1B}$-ATPase were only available for a single inward-open conformation of hATP7B and of AfCopA, representing snapshots of early ion-binding[26,33]. However, AfCopA and hATP7B provided rather different metal interaction models. The M1 methionine (equivalent to M352 in HMA4) at the MB' platform and the two central cysteines of the CPC-motif in M4 (C597 and C599) formed a metal-binding site in AfCopA that demonstrated considerable flexibility[26]. Conversely, a site more buried into the M-domain was proposed for hATP7B, but the CPC-motif involved in all available models for copper binding to P$_{1B-1}$-ATPases was mutated to SPS for generation of the structures[33]. Thus, a proper copper site cannot be expected and, indeed, there is little experimental support for ion presence in the corresponding cryo-EM maps. Additionally, AfCopA and hATP7B were not structurally determined in alternative conformations, thereby further complicating comparative analyses of ion transfer in the same system.

HMA4 displays an overall similar arrangement of the M-domain as for AfCopA and hATP7B in HMA4$^{apo}$/HMA4$^{Cu}$, indicating that the core of the P$_{1B-1}$-type ATPases likely accepts copper using similar principles among prokaryotic and eukaryotic members, including plants (Supplementary Fig. 21). While the resolution of our E1 structures may obscure the analysis, one of the most noticeable E1 → E2P differences in the M-domain occurs locally at the entry site region, with the CPC-motif being considerably rearranged (Fig. 4A–C and Supplementary Fig. 22). The shifted CPC-region suggests the stretch is highly dynamic, as also observed for the E1-Cu state of AfCopA, which is consistent with a role in uptake, high-affinity binding and release. Accordingly, in contrast to AfCopA, in HMA4$^{apo}$ the cysteines of the CPC-motif point away from the proposed entry site, facing M3, M5, and M6, as if the protein is not yet ready to accept its cargo in the absence of metal (Fig. 4A).

Is then the proposed copper binding occurring in HMA4$^{Cu}$? Low-resolution maps are notoriously difficult for the assignment of ions, as we also indicated for hATP7B. However, we note that, by far, the strongest connecting cryo-EM density between two transmembrane segments HMA4$^{Cu}$ is located between the location of M352, C597, and C599. We interpret this as an ion-binding site, with the CPC cysteines directed towards the entry site M352 (Fig. 4B). Validating this finding, this inter-helix interaction is not present HMA4$^{apo}$, and the ion-binding model is almost identical to that found in AfCopA, where the ion was resolved using anomalous diffraction[26]. Thus, copper is likely bound to HMA4$^{Cu}$ but not HMA4$^{apo}$, as delivered to HMA4$^{Cu}$ from MBD$^{-2}$ via M352 on the remodeled MB' platform. M352, C597 and C599 form a triangle-shaped site, with sulfur-copper distances of 2.7, 2.1 and 2.1 Å to M352, C597 and C599, respectively (Fig. 4B, D and Supplementary Figs. 8–11). Regarding the conundrum of one or two high-affinity binding sites, we note that HMA4$^{Cu}$ is not a high-affinity copper binding conformation and that the identified site rather represents an entry site as also shown for AfCopA. However, the most likely high-affinity binding residues, C597 and C599, and M941 of the MXXXS-motif in M6 are already relatively close to each-other, ranging from 2.1 and 2.7 to 6.0 Å (sulfur-to-copper). Analogously, the residues of the chemically less likely two ion-binding sites model are relatively nearby, 2.1 to 10.2 Å for Site I and 6.0 to 7.8 Å for Site II (sulfur or sidechain oxygen-to-copper). Thus, the high-energy copper binding state with occluded metal may look similar to the E1-Cu configuration locally at the ion-binding site.

Interestingly, it appears as if the shape of the CPC-motif is coupled to the orientation of the residues of the invariant YN-motif and the serine of the MXXXS-motif in HMA4$^{apo/Cu}$ (Supplementary Fig. 1a). In HMA4$^{apo}$ these amino acids interact tightly with the sidechains of the

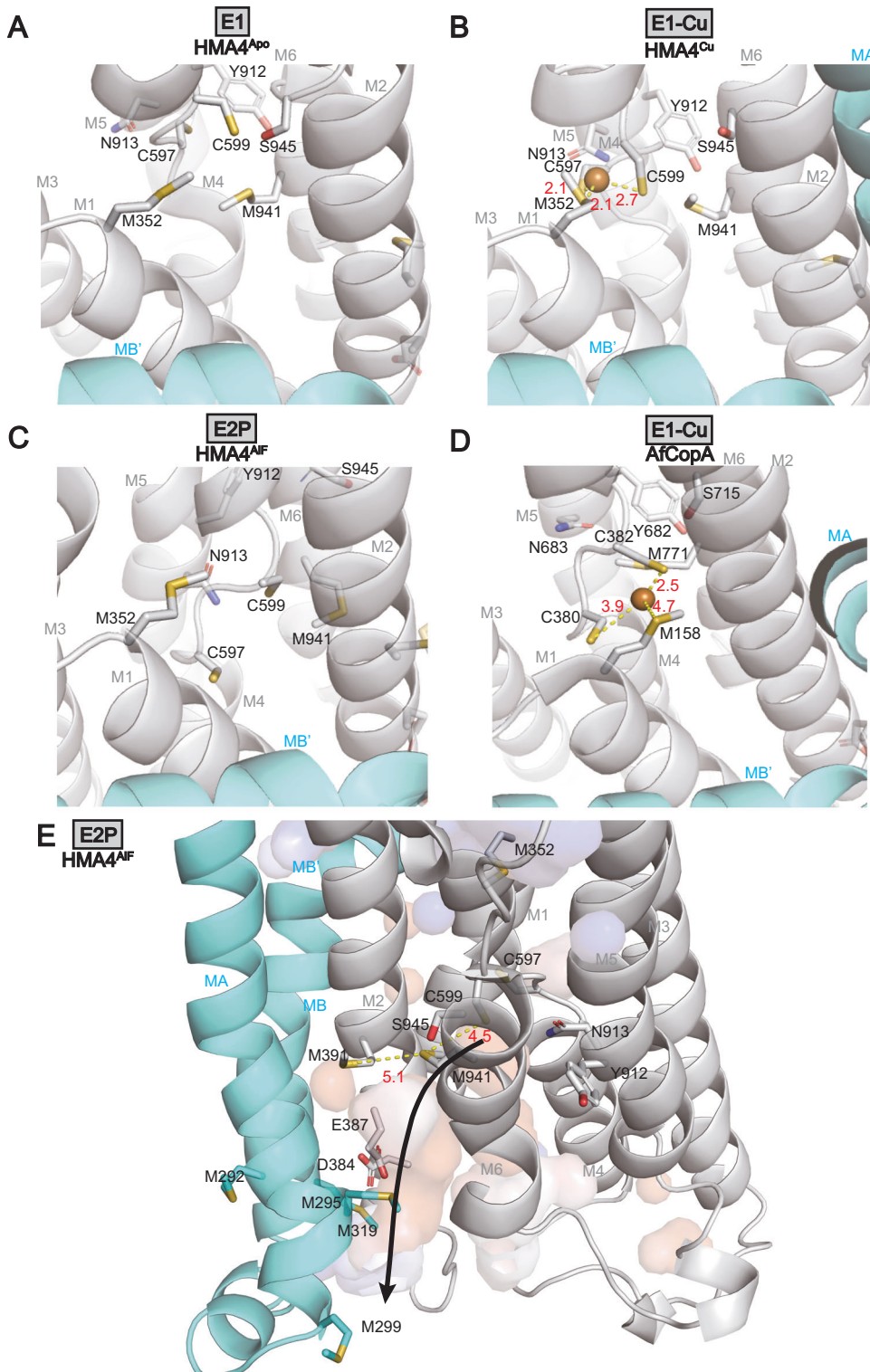

**Fig. 4 | Copper uptake and release.** Supporting cryo-EM density is shown in Supplementary Figs. 8–12 and 22. **A** In the E1 state (HMA4$^{apo}$) the first of the CPC cysteines (C597 of M4) is oriented away, preventing copper uptake. This configuration is stabilized by Y912, N913 (M5) and S945 (M6). **B** In the E1-Cu state (HMA4$^{Cu}$) an entry site for copper is formed, in-between M352 of M1 as well as C597 and C599. **C** Comparison of the copper uptake region in the E2P state (HMA4$^{AlF}$). Note that both the two CPC cysteines are directed away from M352 and hence are incompatible with copper uptake. **D** The equivalent E1-Cu state as in panel b for AfCopA. **E** Copper release in the E2P state (HMA4$^{AlF}$). An opening from the surrounding lumen reaches the CPC cysteines, and it is lined by numerous sulfur-containing residues of MA, M2, and M6, including the conserved M295, M391, and M941 thereby permitting copper release (Supplementary Fig. 1a). The MA link to the soluble domains may represent a point of regulating ion release. Blue and red surfaces denote positively and negatively charged cavities.

central cysteines, with C597 being wedged in-between Y912 and N913, and with a strong interhelical cryo-EM density feature in-between C599 and S945 (Fig. 4A and Supplementary Fig. 10). Conversely, in HMA4$^{Cu}$ the interaction network has altered, with little or no contribution from Y912 and with N913 and S945 rather connecting to the mainchains of C597 and C599, respectively (Fig. 4B). Hence, the YN-motif and the serine of the MXXXS-motif likely manipulate the local environment of the CPC. This observation may explain why these residues have been shown to be critical for function and even affect copper binding, although our data suggest the latter effect is indirect[21,22].

## A unifying model for copper release

How cargo could be released from the high-affinity copper binding site detected for P$_{1B-1}$-type ATPases remains debated. Comparisons of the CPC region of HMA4$^{BeF}$/HMA4$^{AlF}$ and HMA4$^{apo}$/HMA4$^{Cu}$ indicate a considerable shift of the entire region, with C595, C599, and M941 relocating from being relatively close and pointing towards a common center in HMA4$^{Cu}$, to more scattered positions in HMA4$^{BeF}$/HMA4$^{AlF}$ (Fig. 4E). Importantly, in the latter structures C597 and C599 point in opposite directions, towards MA and M6, respectively. Notably, C599 and M941 almost converge, with M941 rearranged to approach M391 of M2, as also orchestrated by the YN-motif. Collectively, these changes must have considerable impact and lower the copper binding capacity of the area, making back-transfer to the cytosol via the uptake region unlikely since all the three ligands are directed away (C599 and M941) or have become more buried (C597). Moreover, the initial transfer from the high-affinity site likely involves a passage dependent on C599 and M941.

How is copper then transferred to the extracellular side? A narrow pathway, lined by MA, M2, and M6, and partially exposed to the surrounding environment was observed in the E2P conformation of LpCopA[23]. In contrast, a wide negatively charged two-legged cavity was detected in AlF-stabilized XtATP7B, lined on one side by M2, M4, and M6 and on the other by MA, MB, M2, and M6[24]. In HMA4$^{BeF}$/HMA4$^{AlF}$ an outward-open, water-accessible conduit reaches the CPC motif cysteines (Fig. 4E). From there and to the extracellular side, the pathway is lined by residues of MA, M2 and M6: including M941, M391, M292, E387, M295, D384 and M319, of which M295, E387, M391, and M941 are highly conserved (Supplementary Fig. 1a). Notably, the sulfur-to-sulfur distances are directly compatible with copper transfer, such as C599 to M941 (4.6 Å) and M941 to M391 (5.6 Å), while the amino acids closer to the outside would rely on sidechain shifts to permit transport, such as M391 to M292 and M295 at 9.6 and 8.4 Å, respectively. Such a reorganization can, however, be accommodated within the determined HMA4$^{BeF}$/HMA4$^{AlF}$ structures. Hence, copper passage entirely through sulfur ligands seems possible, which is important as Cu$^+$ is known to preferably interact with sulfur ligands, although it cannot be excluded that E387 is involved in the process as previously indicated for the equivalent residue of LpCopA[23,25]. As such, our findings shed light also on copper release, and the identified pathway unifies aspects found in both LpCopA and XtATP7B. It is less exposed to the membrane environment than suggested for LpCopA, and it is likely more narrow and not dependent on the two-legs pathway observed in XtATP7B. Interestingly, with the aforementioned link to the soluble domains via MA, turn-over of the soluble domains may as such also assist controlling opening of MA and hence copper release (Fig. 4E).

## Transport and regulation model

We propose that MBD$^{-1}$, which essentially is omnipresent among P$_{1B-1}$-ATPases, in most or all eukaryotic members serves as an N-terminal extension of the A-domain, interacting with it throughout the cycle, and reshaping the ion-uptake region (Fig. 5). This aspect may be essential in vivo. In the absence of copper, most P$_{1B-1}$-ATPases likely maintain an E1 conformation, rendering the M-domain accessible from the cytosol. However, MBD$^{-3}$ and preceding MBDs (if present) may hinder turnover of the soluble domains through interference at the A-/P-domain interface, which also places MBD$^{-2}$ at the MB' platform, thus blocking cargo uptake. We speculate an increasing number of MBDs may increase the copper sensitivity of the autoinhibitory part of the N-terminus. In the presence of copper and ATP, MBD$^{-3}$ and MBD$^{-2}$ become more dynamic and release from the ATPase core, the former in accordance with the release of autoinhibition. This flexibility allows for copper loading via cytoplasmic chaperones such as ATOX1 in human, which delivers the metal to MBD$^{-2}$ or is guided to the ATPase core by the domain, thereby increasing the local concentration of copper available for the pump. Copper delivery independent of MBD$^{-2}$ directly to the MB' platform also from chaperones cannot be excluded. Copper from MBD$^{-2}$ or ATOX1-like chaperones is accepted by the conserved Met of M1 (M352). This methionine forms an entry site together with the CPC cysteines of M4 (C597 and C599), as allowed by a reorientation of the sulfur-containing residues, remodeling of which is stimulated by a copper-induced shift of the YN-motif and the serine of the MXXXS-motif. Once copper has been transferred, MBD$^{-2}$ is released from the MB' platform. We speculate the MBD$^{-2}$ of eukaryotic P$_{1B-1}$-ATPases is replaced by MBD$^{-1}$ in prokaryotic members, as supported by AlphaFold models of e.g. AfCopA, and hence that the MBD$^{-1}$ regulation is not present in lower organisms. As the cytosolic domains shift and are phosphorylated, entering the E1P state, it is possible that substantial local conformational changes are not required for the formation of the following high-affinity site, consisting of the CPC cysteines and the invariant Met of M6 (C597-C599-M941). With the shift to E2P, the TGE-loop of the A-domain moves in for dephosphorylation. On the opposite side of the membrane, the M-domain shifts to outward-open. The residues of the high-affinity binding site (C597-C599-M941) separate, thereby lowering copper affinity and enabling further shuttling down the transport pathway through a chain of sulfur-presenting amino acids lining a pathway within the M-domain. The first step occurs via the second cysteine of the CPC-motif (C599) and the Met of M6 (M941) that interact in E2P. Next, the A-domain dephosphorylates the P-domain, re-occluding the M-domain with the shift to E2. The movements of the cytosolic and transmembrane domains re-enable another transport round, returning the M-domain to an inward-open configuration, providing the opportunity for MBD$^{-3}$ to rebind and inhibit the transport activity in case copper levels are low. While this transport mechanism would overall be the same across species, it would appear that the regulation by MBD$^{-6}$ to MBD$^{-3}$ is species specific, as revealed by the varied number of MBDs, while we expect a more conserved function for MBD$^{-1}$ (in eukaryotes) and MBD$^{-2}$. Structures of the E1P and E2 states are required to further dissect how regulation and high-affinity binding occurs (and if counter-ions are present) in P$_{1B-1}$-ATPases.

## Summary and implications beyond basic protein function

In conclusion, we here present structures of a P$_{1B-1}$-ATPase model protein from rice, HMA4, in the inward-open copper-free (apo) and -bound E1 conformations as well as in the outward open E2P configuration. The arrangement of the A-domain is unique in the E1 state compared to other P-type ATPase classes, and yet it undergoes a considerable rearrangement with the shift to the E2P conformation. The structures shed light on the molecular underpinnings of P$_{1B-1}$-ATPase facilitated copper transport across cellular membranes, revealing aspects related to ion-uptake and -release. Moreover, we show the metal binding domains likely serve triple functional roles. MBD$^{-1}$ is critical, operating as an A-domain extension that remodels the ion uptake region. The position of MBD$^{-2}$ is E1/E2P-dependent and it can likely directly or indirectly assist in donating copper to the MB' platform. MBD$^{-3}$ (and beyond) may be autoinhibitory, sterically hindering turnover of the ATPase core. Notably, our suggestion that

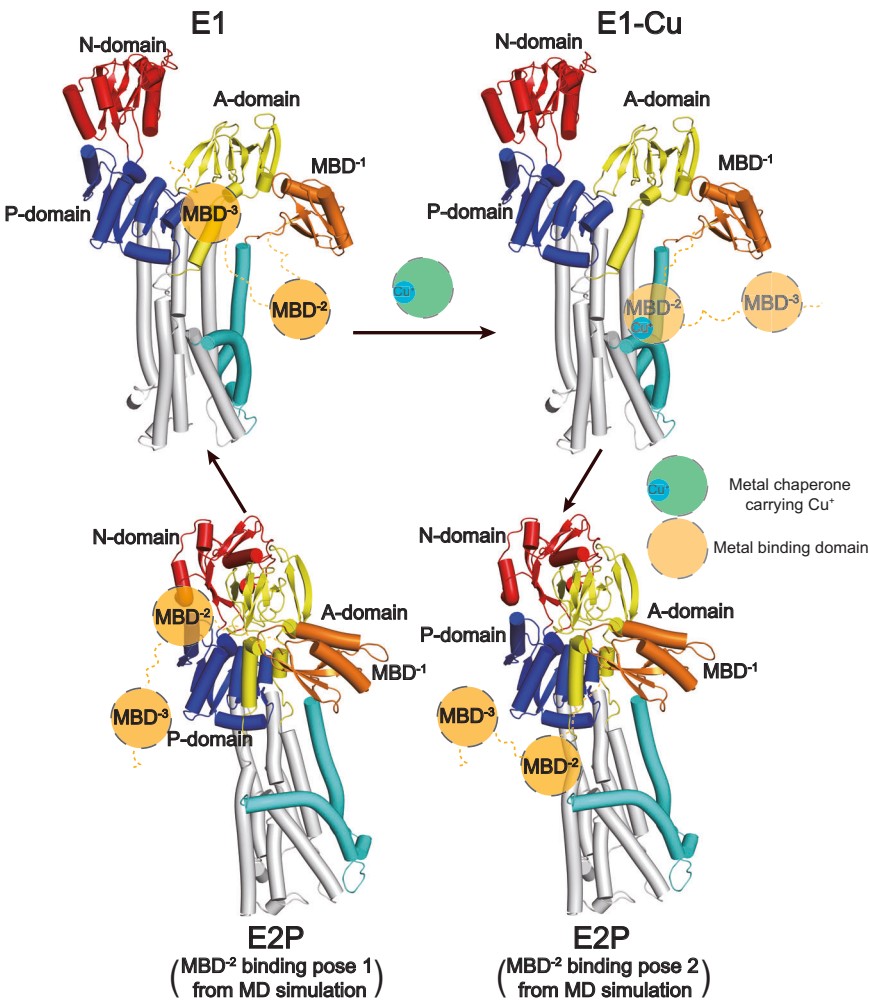

**Fig. 5 | Proposed transport and regulation mechanism of copper-specific P₁B-ATPases.** MBD$^{-1}$ is maintained linked to the A-domain throughout the E1-E1P-E2P-E2 transport cycle. The E1 state is inward-open and capable of binding MBD$^{-3}$. At low copper concentrations, the state can likely be inhibited by MBD$^{-3}$ through prevention of turn-over of the soluble domains. At elevated copper level, MBD$^{-3}$ releases, copper is donated from soluble chaperones to MBD$^{-2}$, and MBD$^{-2}$ docks to the MB' platform for ion delivery. This allows transfer of the metal to the conserved Met of M1 at the MB' platform, and then to the cysteines of the CPC-motif (M4), thereby forming the entry site as observed in the E1-Cu intermediate. Conformational changes from E1-Cu to E1P establishes the high-affinity binding site consistent with the cysteines of the CPC-motif and the Met of the MXXXS-motif. This triggers accomplishment of the E2P form in which metal is shuttled through a pathway lined by MA, M2, and M6 via sulfur-exposing residues. MBD$^{-2}$ can bind to this conformation but likely without inhibitory effect. Following copper release, the protein reverts to the E2 conformation from which a new cycle can be commenced.

MBD$^{-2}$ and, in particular, MBD$^{-1}$ are more important for the function than the preceding metal binding domains resonate with two ongoing clinical studies and previous preclinical trials with ambitions to cure Wilsons disease in which an ATP7B protein lacking MBD$^{-6}$ to MBD$^{-3}$ (VTX-801) or MBD$^{-6}$ to MBD$^{-4}$ (UX701) but with MBD$^{-2}$ and MBD$^{-1}$ retained is re-introduced into the patients[58,59]. We expect that such ATP7B forms will have maintained copper transport capacity as also shown in animal experiments, but at the same time may lack fine-tuned regulation, and that consequently long-term surveillance of treated patients may be required[60]. Our findings can also be exploited for the design of plant copper pumps with MBD$^{-1}$ and perhaps MBD$^{-2}$ for enrichment of copper in edible plants such as rice or accumulation of the metal for removal from contaminated soils.

## Methods
### Primers
The primers for the generation of the employed protein forms were:

Wild-type HMA4:

5'ACACAAATACACACACTAAATTACCGGATCAATTCTAAGATAAT
TATGGAGCAGAATGGAGAGAACCATCTCAAAG3'

5'AAATTGACTTTGAAAATACAAATTTTCCACCAAATCCGGGTCAT
TCTTGGGTC3'

Δ(N-term)HMA4:

5'ACACAAATACACACACTAAATTACCGGATCAATTCTAAGATAAT
TATGAAGACGAGGAAGGTCATG3'

5'AAATTGACTTTGAAAATACAAATTTTCCACCAAATCCGGGTCAT
TCTTGGGTC3'

Δ(MBD$^{-3}$)HMA4:

5'ACACAAATACACACACTAAATTACCGGATCAATTCTAAGATAAT
TATGGAACAGGAAATTGCTGTATGC3'

5'AAATTGACTTTGAAAATACAAATTTTCCACCAAATCCGGGTCAT
TCTTGGGTC3'

Δ(MBD$^{-3}$-MBD$^{-2}$)HMA4:

5'ACACAAATACACACACTAAATTACCGGATCAATTCTAAGATAAT
TATGGATGTGAACAAAGTCCACC3'

5'AAATTGACTTTGAAAATACAAATTTTCCACCAAATCCGGGTCAT
TCTTGGGTC3'

Δ(MBD$^{-3}$-MBD$^{-1}$)HMA4:

5'ACACAAATACACACACTAAATTACCGGATCAATTCTAAGATAAT
TATGTCACCACCAAAGCAAAGAGAG3'

5'AAATTGACTTTGAAAATACAAATTTTCCACCAAATCCGGGTCAT
TCTTGGGTC3'

Wild-type ATP7B:

5'ACACAAATACACACACTAAATTACCGGATCAATTCTAAGATAAT
TATGCCAGAACAAGAAAGACAAATC3'

5'AAATTGACTTTGAAAATACAAATTTTCAATGTATTGTTCTTCAT
CTCTAC3'

Δ(MBD⁻⁶-MBD⁻²)ATP7B:

5'ACACAAATACACACACTAAATTACCGGATCAATTCTAAGATAAT
TATGTCTGATGGTAACATTGAATTGAC3'

5'AAATTGACTTTGAAAATACAAATTTTCAATGTATTGTTCTTCAT
CTCTAC3'

Δ(MBD⁻⁶-MBD⁻¹)ATP7B:

5'ACACAAATACACACACTAAATTACCGGATCAATTCTAAGATAAT
TATGAACGCTCACCACTTGGACC3'

5'AAATTGACTTTGAAAATACAAATTTTCAATGTATTGTTCTTCAT
CTCTAC3'

## Overproduction and purification of HMA4

The expression plasmid for full-length OsHMA4 was generated by in vivo homologous recombination in *Saccharomyces cerevisiae* by transforming the PAP1500 strain[61] with an HMA4 PCR fragment, a TEV-GFP-His$_{10}$ PCR fragment and the expression vector pEMBLyex4 (*BamHI* and *Hin*dIII digested)[62]. Protein production was performed essentially as described previously[63]. Briefly, PAP1500 transformants were selected on synthetic minimal (SD) plates with lysine (30 mg/L) and leucine (60 mg/L). The transformed cells were first grown in 5 mL SD media supplemented with glucose (20 g/L), lysine and leucine at 30 °C, followed by several rounds of up-scaling with leucine starvation to ensure an ultra-high copy number. Finally, a 2 L culture was inoculated with the pre-culture, following procedures already reported[63]. The protein production level was followed by fluorescence microscopy, and yeast cells were typically harvested 96 h following induction. Harvesting and all subsequent steps were performed at 4 °C unless otherwise stated.

20 g of cells (wet-weight) were resuspended in a final volume of 40 mL lysis buffer (20 mM imidazole, pH=7.5, 800 mM NaCl, 10% v/v glycerol, 5 mM β-mercaptoethanol, 1 mM phenylmethylsulfonyl fluoride (PMSF), 1 mM MgCl$_2$ and one SIGMA*FAST* protease inhibitor cocktail tablet/200 mL). Cells were broken with a BeadBeater system from BioSpec using 8 rounds of 1 min high-speed whirl mixing followed by 3 min break. The resulting crude extract was recovered and centrifuged at 2800 g for 20 min, and membranes were collected through a second centrifugation of the supernatant at 185,500 g for 3 h. The resulting membrane pellet was resuspended and homogenized in 20 mL lysis buffer. The membranes were diluted to a final protein concentration of 10 mg/mL in lysis buffer with a final concentration of 2 w/v % solgrade n-Dodecyl-β-D-Maltopyranoside (DDM, Anatrace) and 0.66 w/v % cholesteryl hemisuccinate tris salt (CHS), and stirred for 16 h at 4 °C. The solution was then centrifuged at 140,000 g for 1 h to remove insoluble material. The solution was passed through 2 mL Ni-NTA resin (Thermo Scientific), and the resin was then washed with buffer A (50 mM Tris, pH=7.5, 500 mM NaCl, 10 v/v % glycerol, 30 mM imidazole and 0.03 w/v % DDM, Anatrace) and buffer B (50 mM Tris, pH=7.5, 300 mM NaCl, 10 v/v % glycerol, 30 mM imidazole and 0.05 w/v % lauryl maltose neopentyl glycol (LMNG)). To remove the GFP-His$_{10}$ tag, Tobacco Etch Virus (TEV) protease[64] in a ratio of 1:10 (w/w, TEV protease/protein) was added and incubated for 16 h. Flow-through solution containing GFP-free HMA4 and contaminants was collected. Finally, the protein was concentrated using Vivaspin 20 concentrators (MWCO = 100 kDa) and applied in a concentration of 1 mg/mL to a Superdex 200 increase 10/300 size-exclusion column (Cytiva) equilibrated in buffer C (20 mM Tris-HCl, pH=7.5, 150 mM NaCl, 0.00075 w/v % LMNG, 0.00025 w/v % GDN, 0.00001 w/v % CHS). The fractions containing the protein were pooled and concentrated to 1 mg/mL for cryo-EM sample preparation.

## Cryo-EM structure determination

Purified HMA4 was frozen on Quantifoil 1.2/1.3 holy carbon Au grids which were glow-discharged using a Leica Coater ACE 200 for 60 s with 10 mA current. The grids were prepared using a Vitrobot Mark IV operated at 100 % humidity and 4 °C. In total, 3 µL of purified protein was applied to each grid, incubated for 5 s, blotted for 3 s, and then plunge frozen into liquid ethane. Frozen grids were stored in liquid nitrogen until data collection. The protein in E2 conformations were stabilized using supplements added during TEV protease treatment and size-exclusion chromatography purification: HMA$^{BeF}$ (1 mM BeCl, 1 mM MgCl$_2$ and 3 mM NaF); HMA$^{AlF}$ (1 mM AlCl$_3$, 1 mM MgCl$_2$ and 4 mM NaF). The protein in E1 states were exposed to 0.5 mM CuCl$_2$ and 1 mM tris-(2-chloroethyl)phosphate (E1-Cu) for 1 min before freezing or no such supplementation (E1-apo).

The cryo-EM datasets were collected on two separate Titan Krios electron microscopes (FEI) operated at 300 kV with either a Falcon3 detector in counting mode or a Gatan K3 detector in counting mode and using EPU 3.6. For the Falcon3 dataset, the pixel size was set to 0.832 Å and the total dose was 40 e/Å2 in 40 frames. For the Gatan K3 datasets, the pixel size was 0.8617 Å and the total dose was 50 e/Å2 in 40 frames. An energy filter at 20 eV was applied for the Gatan K3 data collection. All data sets were processed using cryosparc following the procedures outlined in Supplementary Figs. 3–6 and Supplementary Table 1. The data were initially processed using full-frame motion correction and patch CTF determination. Particles with a diameter of 80–120 Å were picked without templates and extracted to a box size of 360 pixels (300 Å diameter) using local motion correction with dose-weighting. Extracted particles were subjected to several rounds of reference-free two-dimensional class averaging to remove obvious junk and contaminations. The cleaned particle set was processed following standard cryosparc (cryosparc v3.3.1) workflow steps including ab-initio model reconstitution, multiple rounds of heterogeneous refinement, and non-uniform refinement iterations.

The initial models were obtained from AlphaFold. The full-length models were fitted into the separate density maps using molecular dynamics flexible fitting (MDFF of NAMD) in an implicit solvent with secondary structure, cis-peptide, and chirality restraints[65]. The models were further refined by multiple rounds of real-space refinement in Phenix[66] and manual adjustment in coot[67]. Model building and refinement were performed using USCF chimera 1.14, NAMD v2.14, Wincoot 0.9.2 and Phenix 1.20.1. Figures were generated using USCF chimera v1.14, ChimeraX v1.5 and pymol v2.1. Sequence alignments were performed using Cluster Omega (online), and visualized using Jalview v2.11.2.7.

## Overproduction and purification of ATP7B variants

Full-length human ATP7B and truncated ATP7B versions were heterogeneously overproduced and membranes were prepared as described above. The membranes were diluted to a final protein concentration of 3 mg/mL in lysis buffer with a final concentration of 1 w/v % DDM and 0.1 w/v % CHS, and incubated with stirring for 2 h at 4 °C. The solution was then centrifuged at 140,000 g for 1 h to remove insoluble material. Solid KCl and imidazole were added to the solubilized membranes to a final concentration of 500 mM and 50 mM, respectively. The solution was passed through a 5 mL His-trap column using an ÄKTA pure system, followed by elution of the tagged protein with a gradient of buffer D (20 mM Tris-HCl pH=7.5, 200 mM KCl, 20 v/v % glycerol, 5 mM β-mercaptoethanol, 1 mM MgCl$_2$, 0.15 mg/mL C$_{12}$E$_8$ and 0.015 mg/mL CHS) containing 500 mM imidazole. To remove the GFP-His$_{10}$ tag the protein was dialyzed for 16 h in buffer E (20 mM Tris-HCl, pH=7.5, 80 mM KCl, 20 v/v % glycerol, 5 mM β-mercaptoethanol, 1 mM MgCl$_2$, 0.15 mg/mL C$_{12}$E$_8$ and 0.015 mg/mL CHS) in the presence of TEV protease in a ratio of 1:10 (w/w, TEV/protein). A second immobilized metal ion affinity chromatography round was applied to remove non-cleaved protein, free GFP-His$_{10}$ and remaining

contaminants. Finally, the protein was concentrated using Vivaspin 20 concentrators (MWCO = 100 kDa) and applied in a concentration of 10–15 mg/mL to a Superose 6 size exclusion column (Cytiva) equilibrated in buffer C (20 mM Tris-HCl, pH=7.5, 80 mM KCl, 20 v/v % glycerol, 5 mM β-mercaptoethanol, 3 mM $MgCl_2$, 0.15 mg/mL $C_{12}E_8$ and 0.015 mg/mL CHS). The fractions containing the protein were pooled and concentrated to 10 mg/mL.

## Activity measurements
The activity of ATP7B was measured using the Baginski method[44]. 4 μM protein was added to the reaction buffer (40 mM MOPS-KOH, pH=6.8, 150 mM NaCl, 5 mM KCl, 5 mM $MgCl_2$, 1.2 mg/mL L-α-phosphatidylcholine from soybean, 3.7 mM $C_{12}E_8$, 10 mM cysteine, 5 mM Tris(2-carboxyethyl)phosphine and 100 μM $CuCl_2$), and to exclude possible stimulation from minor contaminations of other ATPases also azide (5 mM, final), nitrate (20 mM) and molybdate (0.2 mM) was included in the reaction buffer, inhibiting $F_1F_0$-ATPases[68], V-ATPases[69] and soluble acid phosphatases[70]. The reaction was started through the addition of 5 mM ATP, and at 5–10 min intervals 50 μL of the assay solution was transferred to a 96-well microplate and mixed with an equal volume of ascorbic acid solution (formed by mixing a solution containing 0.17 M ascorbic acid and 0.1% SDS in 0.5 M HCl with aqueous 28.3 mM ammonium heptamolybdate in a 5:1 ratio). After incubation for 10 min at 18 °C using 75 uL of sodium arsenic solution (0.068 M trisodium citrate, 0.154 M sodium metaarsenic, 2 v/v % glacial acetic acid) was added, and the absorbance measured at 860 nm following 30 min.

## Yeast complementation assay
The $ccc2\Delta$ strain PAP6064 (*mat a his3 Δ1::UPR-lacZ HIS3 leu2Ο met15 ΔΟ ura3υ ΔΟ ccc2::kanMx4*) was used as host for the complementation assay. Each $P_{1B-1}$-ATPase allele was expressed from the high copy number plasmid pEMBLyex4 as described previously[61]. Complementation was analyzed by spotting 5 μL of exponentially growing cells with optical density (OD$_{450nm}$) = 0.5, 0.05 and 0.005 onto SG (minimal medium with galactose) plates with 1 mM Ferrozine, 1 μM $CuCl_2$ and 135 μM Fe(NH$_4$)$_2$SO$_4$. To confirm viability, the same amount of each yeast culture was also spotted on non-selective medium (SG minimal medium). The plates were incubated at 30 °C and inspected daily.

## Estimation of recombinant copper ATPase expression levels
GFP fluorescence from PAP6064 cells producing either no copper ATPase, hATP7B-GFP, wild type HMA4-GFP or HMA4-GFP mutants was used for relative quantification of expression levels. Cells were grown at 30 °C for 24 h in minimal medium with 2 % galactose as carbon source. One OD$_{450}$ unit of cells was harvested, re-suspended in 200 μL $H_2O$ and transferred to the wells of a white microplate and fluorescence was subsequently measured in a microplate spectrofluorometer (Fluoroskan Ascent, Thermo Labsystems) using 485 nm excitation and 520 nm emission.

## Live cell bioimaging
The *S. cerevisiae* cells used for estimation of expression levels were used for live cell bioimaging too. Fluorescence was visualized at ×1000 magnification with a Nikon Eclipse E600 fluorescence microscope equipped with an Optronics Magnafire model S99802 camera.

## Molecular dynamics simulations
Three atomic models covering the HMA4 core domains and MBD$^{-3}$ to MBD$^{-1}$ were built based on the experimental structures and an Alpha-Fold model of HMA4[71]. In all models MBD$^{-1}$ was placed as in the experimental structures, and MBD$^{-3}$ was positioned away from the core domains. The E1 model (E1$^{apo}$) was based on HMA4$^{apo}$ and has MBD2 in the position predicted by AlphaFold. The two E2P models were based on HMA4$^{AlF}$ and has MBD$^{-2}$ in the position predicted by

AlphaFold of HMA4 (E2P$^{AlF}$) and the position seen in the structure of XtATP7B (E2P$^{Xt}$)[24], respectively. The models were inserted into 1-palmitoyl-2-oleoyl-sn-*glycero*-3-phosphocholine (POPC) lipid membranes using the CHARMM-GUI[72,73] Membrane Builder[74,75] and solvated with TIP3P water[76] with 150 mM KCl. Residue E723 was protonated in E1$^{apo}$ as predicted by PROPKA3.1[77,78]. Ten independent replicas were simulated for the E1$^{apo}$ system and five independent replicas were simulated for the E2P$^{AlF}$ and E2P$^{Xt}$ systems (Supplementary Table 3). Each system was energy minimized with a steepest descent algorithm until Fmax <1000 and equilibrated with a stepwise release of position restraints on the protein and lipids for a total of 1.875 ns. Position restraints from the last equilibration step were kept on the backbone atoms of the cryo-EM structures for the production simulations while MBD$^{-3}$, MBD$^{-2}$ and the inter-MBD linkers were free to move. The equilibrated structures were simulated for 200 ns with a v-rescale thermostat at 298.15 K[79] and a berendsen semiisotropic barostat at 1 bar with a compressibility of 4.5e −5 bar$^{-1}$[80]. A 12 Å cut-off was used for non-bonded interactions. Because the biological time scales of MBD rearrangements are out of range for brute-force simulation, we explored MBD dynamics in the context of a restrained protein backbone until RMSD differences were obtained (Supplementary Fig. 19 and Supplementary Table 4). All simulations and RMSD calculations were performed with the GROMACS-2021 simulation package[81] using the charmm36m all-atom force field[82] for proteins and the C36 lipids[83] forcefield for lipids.

## Data availability
The data that support this study are available from the corresponding authors upon request. Cryo-EM maps have been deposited in the Electron Microscopy Data Bank (EMDB) under accession codes EMD-18202 (E1 - HMA4$^{apo}$), EMD-18203 (E1-Cu - HMA4$^{Cu}$), EMD-18205 (E2P - HMA4$^{BeF}$) and EMD-18204 (E2P - HMA4$^{AlF}$). The atomic coordinates have been deposited in the Protein Data Bank (PDB) under accession codes 8Q73 (E1 - HMA4$^{apo}$), 8Q74 (E1-Cu - HMA4$^{Cu}$), 8Q76 (E2P - HMA4$^{BeF}$) and 8Q75 (E2P - HMA4$^{AlF}$). All files required for producing the trajectories of the MD simulations can be found at repository: https://doi.org/10.5281/zenodo.10708625. Source data are provided with this paper.

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

## Acknowledgements

This study was supported by The Lundbeck Foundation (R322-2019-2588; V.B., R133-A12689; K.W., R133-A12689, R218-2016-1548, R313-2019-774 and R346-2020-2019; P.G.), The Novo Nordisk Foundation (NNF16OC0021272 and 0078574; P.G.), the Danish Council for Independent Research (6108-00479 and 9039-00273; P.G.), the Swedish Research Council (2020-03840; M.A., 2016-04474 and 2022-01315; P.G.), Knut and Alice Wallenberg Foundation (2015.0131 and 2020.0194; P.G.), the Crafoord Foundation (20170818, 20180652 and 20200739; P.G.), The Carlsberg Foundation (CF15-0542 and CF21-0647; P.G.), the Per-Eric and Ulla Schybergs Foundation (38267; P.G.), the Augustinus Foundation for equipment (16-1992; P.G.,), the Brødrene Hartmanns Foundation (A29519; P.G.), the Agnes and Poul Friis Foundation (n/a; P.G.), as well as by the Japan Society for the Promotion of Science (21H05034; J.F.M.). The post-doc fellowship of CG was supported by The BRIDGE - Translational Excellence Programme at University of Copenhagen funded by the Novo Nordisk Foundation. C.G. was also financially assisted by The memorial foundation of manufacturer Vilhelm Pedersen and wife—and the Aarhus Wilson consortium. Computational resources were provided by the Swedish National Infrastructure for Computing (SNIC) through the High-Performance Computing Center North (HPC2N) under project SNIC 2022/5-168 and by the High-Performance Computing Center North (HPC2N) under project hpc2n2023-005. The funders had no role in study design, data collection and analysis, decision to publish, or preparation of the manuscript. We would like to thank Tillmann Hanns Pape at the Danish Cryo-EM Facility at the Core Facility for Integrated Microscopy (CFIM) at University of Copenhagen for assistance with sample screening and data collection. The Danish Cryo-EM Facility at CFIM, University of Copenhagen is supported by Novo-Nordisk Foundation grant no. NNF14CC0001. We would also like to thank Julian Conrad, Karin Wallden, Dustin Morado and Marta Carroni at the Cryo-EM Swedish National Facility in Stocholm for sample screening and data collection. The Cryo-EM Swedish National Facility at SciLifeLab is funded by the Knut and Alice Wallenberg, Family Erling Persson and Kempe Foundations, SciLifeLab, Stockholm University and Umeå University.

## Author contributions

M.A., P.A.P., K.W., and P.G. supervised the project. J.F.M. provided the HMA4 clone. Z.G. designed the experiments to establish protein overproduction, purification for the cryo-EM structural studies. K.W.

prepared the grids for the generation of the structures, and K.W. collected the data for HMA4apo and HMA4AlF at CFIM of the University of Copenhagen. The HMA4Cu and HMA4BeF data were gathered at SciLife-Lab in Stockholm, assisted by Z.G. Z.G. processed the cryo-EM datasets, built and refined the structures. The MDFF for HMA4apo was accomplished by Y.W. and then by Z.G. for the other three structures. F.O. performed the MD simulations, assisted by M.A. P.A.P. provided protein for the functional studies of hATP7B instructed Z.G. for the production of HMA4, and performed the bioimaging, the complementation assay as well as the determination of the expression levels of the various $Cu^+$-ATPase variants. C.G. conducted the purification and functional studies of hATP7B. Z.G. prepared the figures, except for the ones related to the functional characterization of hATP7B (C.G.) and the distance difference matrix (V.B.). Z.G. and P.G. conducted the initial data analysis and wrote the first draft. P.O. assisted with the interpretation of the medical implications of the results. All authors conducted data analysis and interpretation. Z.G., V.B., and P.G. finalized the manuscript. All authors commented on the manuscript.

## Funding

## Competing interests

The authors declare no competing interests.
