## [Peer Review File · Nature Communications]

Diverse roles of the metal binding domains and transport mechanism of copper transporting P-type ATPasesReviewer #1 (Remarks to the Author):

In this manuscript, Guo and colleagues determined 4 new structures of the copper transporting ATPase HMA4 from plants. The structures describe the apo-protein, the copper bound protein, and two inhibitors-stabilized phosphorylation intermediates. The structural models have reasonable resolution allowing the authors to discuss several new aspects of the copper-transport mechanism. The previously published different conformers were obtained for different species (frog and human), and therefore it is important that in this work the conformers are obtained and analyzed for the same copper pump. Another significant finding is the direct evidence for the structural role of the metal binding domain nearest to the membrane domain (MBD-1). The lack of the CxxC motif rules out the previously proposed role of this domain in Cu delivery to the membrane site. Another interesting and new finding is the distinct position of the A-domain that seems to be determined by the linker. There is also nice evidence suggesting that the YN-motif and the Ser residue of the MXXXS-motif may modulate the environment of the CPC motif.

Despite these good and solid data, the writing style does significant disservice to the authors. By constantly mixing the discussion of the previously reported structures, the AlphaFold model, and their own results, the authors confuse rather than help the reader. It is difficult to separate what are the truly new and firm findings, what was already reported in previous studies, and what the authors are proposing/hypothesizing based on their comparisons with the available data. Making these points clearer and describing their data first would help the authors to highlight the novelty and significance of their work. At the moment, as written, the work does not sound particularly novel.

The section "MBD-2 likely delivers copper to the entry site" is speculative and does not have strong experimental support

It is also unclear and confusing why the functional data are reported for human ATP7B rather than for HMA4. This makes little sense

Minor comments

Lane 51. Dysfunctional hATP7A and hATP7B is... replace "is" with "are"

lane 203 "...a peculiar arrangement of the A-domain" – the authors should spell out what they mean by that

This reviewer would argue that the experimental data should serve as a verification of the AlphaFold predictions and not the other way around. The discussion of the AlphaFold model should not go beyond highlighting the agreements (or disagreements) with the actual experimental data.

Reviewer #2 (Remarks to the Author):

Guo et al analyzed cryo-EM structures of Cu-transporting P1B ATPase from rice including apo state, Cu-bound E1 state and Cu-extruded E2P states, revealing conformational changes during the transport cycle. As has previously shown in XtATP7B, authors determined MBD-1 is attached to the A domain, which is clearly different from the hATP7B E1 state in which MBD is assigned at the MB' platform. Using the AlphaFold2 model and MD simulation, the authors proposed that MBD-5 may be bound to the MB' Cu loading platform, which is consistent with the fact that the MBD-1 of HMA4 does not have the CxxC Cu-binding motif.

Experimentally determined cryo-EM structures presented here provide important information regarding the Cu-transport mechanism. However, this paper suffers insufficient functional analysis to support their hypothesis, and therefore discussion only based on the predicted structure and MD

simulation (without supported solid experimental functional analysis) needs to be carefully evaluated. In addition, the paper could be made more accessible to the readers by improving the presentation. References and figures should be cited appropriately in the text. There are some discussions based on AF2 models, but these are mostly not shown in the figure. The atomistic details determined by cryo-EM are not effectively presented in the figure. I hope the following suggestion may improve the manuscript.

Major comments

The results of functional analysis are a fatal flaw for this paper. First of all, as the authors themselves stated in L199-L201, if the results of Cu transporters in various species make them difficult to interpret, authors must evaluate HMA4 activity instead of human ATP7B. Determined structures are Cu transporter from rice, it is natural and mandatory to interpret structures using the same protein. Based on Bitter et al, 2022, Sci Adv, a deletion mutant of XtATP7B containing only MBD-1 and MBD-2 shows 2~3 times higher ATPase activity relative to full-length WT. Therefore, WT ATPase activity shown in this paper is likely autoinhibited by MBD-3 to MBD-6, and thus is underestimated as a maximum ATPase activity. Authors described that deletion mutant with only MBD-1 shows approximately 50% specific activity compared to WT to emphasize the functional significance of this MBD-1 domain. If the maximum (not autoinhibited) ATPase activity is 3 times higher than WT, the ATPase activity of delta(MBD-6~MBD-2) would be only 16.6% of the maximum ATPase activity. This logic is clearly biased, and misleading to the readers. At least, authors should measure delta(MBD-6~MBD-3) mutant to show the potential maximum ATPase activity if they want to emphasize the importance of MBD-1 based on the activity.

Similarly, if MBD-1 is important as an A domain scaffold, its CxxC motif may not be required for the ATPase activity. Evaluation of such a mutant, or citing previous results for such mutants may be helpful.

Two considerations about the functional roles of MBD binding at the A-P domain interface appear in this paper: one is based on the fact that MBD-2 is resolved in the E2P state of XtATP7B (ref 22), and the other is based on the MBD-3 binding observed in the MD simulation of HMA4 by the authors. The former (MBD-2 binding at A-P domain interface) is interpreted that the binding is weak or tentative and does not contribute to the autoinhibition, while the latter, on the contrary, states that it is the cause of autoinhibition, even though no such a structure has been determined experimentally. The authors state that these results explain the previously reported autoinhibition of ATP7B ATPase activity. The ATPase activity measurements for a series of MBDS deletion mutants are needed to draw this conclusion. Comparison with CxxC motif-deficient mutants of each domain would clarify their role (scaffold or Cu delivery).

The description of the atomistic detail of the Cu binding site is interesting and deserves to be described in more detail. There are a few concerns as follows;

Although the EM density is shown in the supplements, it is not clear whether the side chains that directly or indirectly contribute to the Cu binding site, including cysteine residues, are resolved enough to identify their rotamers. The authors should provide a better and clearer expanded figure of the density that supports these models.

L444 It seems difficult to discuss detailed distances between side chain sulfur and Cu, as the EM density is not separated. Besides EM density, is there any analysis to determine if these distances are reasonable? Wouldn't a comparison with the coordination of other Cu-binding proteins be useful?

L455 The description in this paragraph is important to understand conformational changes of the binding site upon Cu binding, but unfortunately it is difficult to follow this, especially the relative special orientations between CPC at M4, and Y912 and N913 at M5, only from the information in Fig. 4.

Minor comments

Fig. 1. Describe how the two models were superimposed. The difference in Fig 1D is particularly larger

than the differences in the other figures, which is confusing for the readers. Comparison of E1 and E2P states from different viewpoints may be helpful to follow its conformational change. Also, in Fig. 1D, A domain in the E2P state is difficult to recognize. For example, the use of light colors corresponding to each domain will help the readers.

L211 Cite Supp Fig 13. It is unclear from this figure that the different position of M3 changes the exposure of the MB platform. Would it be possible to effectively show the different degrees of opening for this area by showing the molecular surface? Also, here the information on how to superimpose these structures is missing, readers (at least for this reviewer) who look at Fig. 1D expect much larger conformational changes in the E1-E2P transition.

L231-232 As far as mentioning the AF2 model, it should be shown and compared to the model in the figures.

L276 The difference in structure does not appear to be enhanced in LpCopA. Rather, the MB of LpCopA is closer to the ATPase core. Structural comparison in Fig. 2E does not seem to be an appropriate example of what the authors want to describe here.

L310 According to Figure 1CD of reference 22, it seems that MBD-2 (MBD5 in the ref) is better resolved than MBD-1 (MBD6). It is not acceptable to say that the interaction is weak only in this area based on the low resolution of the domain.

L323 Show the AlphaFold2 model itself in the figure.

L353 It is unclear what "most of latter" indicates.

L354 Show 8IOY density map as shown in 7XUM in Supplementary Figure 16

L382 MD Data is not shown in the paper.

L386 There is no MBD-3 in Fig 3d

L405 Does Cu-binding to the MBD-3 release the autoinhibition? Is there any evidence or references indicating this? Supplementary Figure 17 only shows that the ATOX1 does not interfere with the ATPase core when previously determined ATOX1-MBD complex structure is superimposed to the assumed structure obtained by the MD run, therefore is not a direct basis for concluding that this causes a release of autoinhibition by MBD-3.

Fig. 3. The Order of displayed panels is confusing, especially for panel F.

L509 Why being E1 without Cu would be supported by the AF2 model?

Figure 3. Please indicate CxxC motif in each MBD domain by sticks or spheres.

Response to reviewers

We thank the reviewers for the evaluation and for the helpful suggestions and comments to improve the manuscript "Diverse roles of the metal binding domains and transport mechanism of copper transporting P-type ATPases". We have now addressed all comments and amended the manuscript as outlined below, with the changes highlighted in yellow in the main manuscript and supplementary material (for new figures the title is highlighted in yellow). Remarks and questions from the reviewers are shown in black. Our responses are shown in red.

Reviewer #1 (Remarks to the Author):

In this manuscript, Guo and colleagues determined 4 new structures of the copper transporting ATPase HMA4 from plants. The structures describe the apo-protein, the copper bound protein, and two inhibitors-stabilized phosphorylation intermediates. The structural models have reasonable resolution allowing the authors to discuss several new aspects of the copper-transport mechanism. The previously published different conformers were obtained for different species (frog and human), and therefore it is important that in this work the conformers are obtained and analyzed for the same copper pump. Another significant finding is the direct evidence for the structural role of the metal binding domain nearest to the membrane domain (MBD-1). The lack of the CxxC motif rules out the previously proposed role of this domain in Cu delivery to the membrane site. Another interesting and new finding is the distinct position of the A-domain that seems to be determined by the linker. There is also nice evidence suggesting that the YN-motif and the Ser residue of the MXXXS-motif may modulate the environment of the CPC motif.

Despite these good and solid data, the writing style does significant disservice to the authors. By constantly mixing the discussion of the previously reported structures, the AlphaFold model, and their own results, the authors confuse rather than help the reader. It is difficult to separate what are the truly new and firm findings, what was already reported in previous studies, and what the authors are proposing/hypothesizing based on their comparisons with the available data. Making these points clearer and describing their data first would help the authors to highlight the novelty and significance of their work. At the moment, as written, the work does not sound particularly novel.

Thank you for the feedback. The entire manuscript has been edited in an attempt to make it clearer. In addition, the MD simulations have been collected in a new section that has been inserted after the initial description of the structures, their states and MBD⁻¹ as well as the functional data, in an attempt to collect all data generated in this paper in one place. An effort has also been made to highlight the novelty of our results and analysis.

The section "MBD-2 likely delivers copper to the entry site" is speculative and does not have strong experimental support

We agree that it is not possible to conclusively assign MBD⁻² to a role in ion delivery. We have endeavored to make this section clearer, and sound somewhat less speculative.

It is also unclear and confusing why the functional data are reported for human ATP7B rather than for HMA4. This makes little sense

We kindly disagree with the reviewer. It is an established practice to study model proteins to learn also how related proteins operate. Indeed, the transport principles across the M-domain of P_{1B}-ATPases appear remarkably conserved from prokaryotes up to the highest eukaryotes, as also discussed in this manuscript.

As we show in the manuscript, the binding and hence functional role of MBD⁻¹ is conserved among many eukaryotes (certainly in HMA4, XtATP7B and hATP7B and well beyond those). Moreover, the brief linker in-between MBD⁻¹ and MBD⁻², experimental structural data (in the E2P state for XtATP7B and in the E1 state for hATP7B), AlphaFold models (typically in E1 states), as well as our MD simulations all point to a dynamic role of MBD⁻², with docking of MBD⁻² placing its CXXC-motif towards the entry site of the ATPase core as a key observation that clearly must be directly linked to its functional role. Thus, it must be anticipated that the roles of both MBD⁻² and MBD⁻¹ are maintained across most eukaryotic species (which is the reason for the adaption of the MBD nomenclature that we suggest), and hence that truncations of these two domains can be studied in complementary eukaryotic studies. The additional benefit with assessing hATP7B is that the data extend beyond the information already available (for example for XtATP7B and the human targets).

Nonetheless, we have now attempted to perform similar functional studies as for hATP7B on HMA4. However, as indicated in the revised manuscript, these have been fruitless, despite extensive efforts. Instead, we provide data generated using HMA4 in a complementation assay, which further supports the different roles we propose for the different MBDs.

Minor comments

Lane 51. Dysfunctional hATP7A and hATP7B is... replace "is" with "are"

This has been corrected.

lane 203 "...a peculiar arrangement of the A-domain" – the authors should spell out what they mean by that

This phrase has been removed and instead this rearrangement has been described.

This reviewer would argue that the experimental data should serve as a verification of the AlphaFold predictions and not the other way around. The discussion of the Alphafold model should not go beyond highlighting the agreements (or disagreements) with the actual experimental data.

We agree with the reviewer and have attempted to adapt the manuscript accordingly.

Reviewer #2 (Remarks to the Author):

Guo et al analyzed cryo-EM structures of Cu-transporting P1B ATPase from rice including apo state, Cu-bound E1 state and Cu-extruded E2P states, revealing conformational changes during the transport cycle. As has previously shown in XtATP7B, authors determined MBD-1 is attached to the A domain, which is clearly different from the hATP7B E1 state in which MBD is assigned at the MB' platform. Using the AlphaFold2 model and MD simulation, the authors proposed that MBD-5 may be bound to the MB' Cu loading platform, which is consistent with the fact that the MBD-1 of HMA4 does not have the CxxC Cu-binding motif. Experimentally determined cryo-EM structures presented here provide important information regarding the Cu-transport mechanism.

We note our reanalysis of the previously published hATP7B data suggest MBD⁻² is located at the MB' platform also for hATP7B in the E1 state, so there is indeed also experimental support for this position.

However, this paper suffers insufficient functional analysis to support their hypothesis, and therefore discussion only based on the predicted structure and MD simulation (without supported solid experimental functional analysis) needs to be carefully evaluated.

We kindly disagree with the reviewer. As we show in the manuscript, the binding and hence functional role of MBD⁻¹ is conserved among many eukaryotes (certainly in HMA4, XtATP7B and hATP7B and well beyond those). Moreover, the brief linker in-between MBD⁻¹ and MBD⁻², experimental structural data (in the E2P state for XtATP7B and in the E1 state for hATP7B), AlphaFold models (typically in E1 states), as well as our MD simulations all point to a dynamic role of MBD⁻², with docking of MBD⁻² placing its CXXC-motif towards the entry site of the ATPase core as a key observation that clearly must be directly linked to its functional role. We have also added data on HMA4 using a complementation assay that add to this analysis.

In addition, the paper could be made more accessible to the readers by improving the presentation. References and figures should be cited appropriately in the text. There are some discussions based on AF2 models, but these are mostly not shown in the figure. The atomistic details determined by cryo-EM are not effectively presented in the figure. I hope the following suggestion may improve the manuscript.

Thank you for the helpful feedback. The paper has been revised overall to attempt to make it clearer. Additional references have been added, where appropriate, and the figure legends have been revised.

Major comments

The results of functional analysis are a fatal flaw for this paper. First of all, as the authors themselves stated in L199-L201, if the results of Cu transporters in various species make them difficult to interpret, authors must evaluate HMA4 activity instead of human ATP7B. Determined structures are Cu transporter from rice, it is natural and mandatory to interpret structures using the same protein.

The reviewer has misunderstood our argument. We do indeed indicate that "the validity of these studies could be questioned considering the spread in origin of the compared members". However, as we and other have shown for P-type ATPases and other targets, this is generally not a concern. Indeed, the transport principles across the M-domain of P_{1B}-ATPases appear remarkably conserved from prokaryotes up to the highest eukaryotes, as also discussed in this manuscript.

Moreover, as we show in the manuscript, the binding and hence functional role of MBD⁻¹ is conserved among many eukaryotes (in HMA4, XtATP7B and hATP7B, and certainly well

beyond those). Moreover, the brief linker in-between MBD⁻¹ and MBD⁻², experimental structural data (in the E2P state for XtATP7B and in the E1 state for hATP7B), AlphaFold models (typically in E1-like states), as well as our MD simulations all point to a dynamic role of MBD⁻², with docking of MBD⁻² placing its CXXC-motif towards the entry site of the ATPase core as a key observation that must be directly linked to its functional role. Thus, it must be anticipated that the roles of both MBD⁻² and MBD⁻¹ are maintained across most eukaryotic species (which is the reason for the adaption of the MBD nomenclature that we suggest), and hence that truncations of these two domains can be studied in complementary eukaryotic studies. The additional benefit with assessing hATP7B is that our data extend beyond the information already available (for example for XtATP7B and the human targets).

Nonetheless, we have now attempted to perform similar functional studies as for hATP7B on HMA4. However, as indicated in the revised manuscript, these have been fruitless, despite extensive efforts. Instead, we provide data generated using a complementation assay, which further supports the different roles we propose for the different MBDs.

Based on Bitter et al, 2022, Sci Adv, a deletion mutant of XtATP7B containing only MBD-1 and MBD-2 shows 2~3 times higher ATPase activity relative to full-length WT. Therefore, WT ATPase activity shown in this paper is likely autoinhibited by MBD-3 to MBD-6, and thus is underestimated as a maximum ATPase activity. Authors described that deletion mutant with only MBD-1 shows approximately 50% specific activity compared to WT to emphasize the functional significance of this MBD-1 domain. If the maximum (not autoinhibited) ATPase activity is 3 times higher than WT, the ATPase activity of delta(MBD-6~MBD-2) would be only 16.6% of the maximum ATPase activity. This logic is clearly biased, and misleading to the readers. At least, authors should measure delta(MBD-6~MBD-3) mutant to show the potential maximum ATPase activity if they want to emphasize the importance of MBD-1 based on the activity.

We disagree with this analysis. The assay employed in by Bitter et al, 2022, Sci Adv, was performed without supplementation of copper (other than from co-purification with the sample), enabling detection of autoinhibition. However, the most likely trigger of release of autoinhibition achieved via copper-sensing domains of a copper-transporter is copper. So, in the presence of copper, as we have supplemented to our functional analysis, autoinhibition is not to be expected.

Nonetheless, delta(MBD⁻⁶-MBD⁻²) serves as a control for the functional role of MBD⁻¹, suggesting that while 'only' serving a structural role, MBD⁻¹ is still important for the function. Moreover, a more significant role on the activity of MBD⁻² (and MBD⁻¹) is certainly consistent with the roles in metal delivery for MBD⁻² (and structural, positioning MBD⁻², for MBD⁻¹), that we are proposing based on our own structural and functional data, our MD simulations, and structural and functional data available for XtATP7B and hATP7B as well as AlphaFold models. So, in the case the reviewer is correct, such data with a more significant difference between wild-type and our truncations would be fully consistent with the proposed mechanistic models of the MBD, in fact further supporting our hypotheses.

Similarly, if MBD-1 is important as an A domain scaffold, its CxxC motif may not be required for the ATPase activity. Evaluation of such a mutant, or citing previous results for such mutants may be helpful.

We agree with the reviewer, as also indicated in the previous and current version of the manuscript. We reason(ed) that evolutionary analysis supports the notion that the CXXC-motif of MBD⁻¹ is insignificant for the ATPase function, with the CXXC-motif of MBD⁻¹ being obsolete in several eukaryotic species (a selection of species are mentioned in the manuscript). That said, we expect copper binding to the CXXC-motif is possible (when present/intact, as have been shown previously) but that this has no effect on the

ATPase function. Future more detailed studies will be needed to further dissect the function of the CXXC-motif of MBD⁻¹; the role of the CXXC-motif of MBD⁻¹ does not represent a key finding of the current study (even missing in HMA4).

Two considerations about the functional roles of MBD binding at the A-P domain interface appear in this paper: one is based on the fact that MBD-2 is resolved in the E2P state of XtATP7B (ref 22), and the other is based on the MBD-3 binding observed in the MD simulation of HMA4 by the authors. The former (MBD-2 binding at A-P domain interface) is interpreted that the binding is weak or tentative and does not contribute to the autoinhibition, while the latter, on the contrary, states that it is the cause of autoinhibition, even though no such a structure has been determined experimentally. The authors state that these results explain the previously reported autoinhibition of ATP7B ATPase activity. The ATPase activity measurements for a series of MBDs deletion mutants are needed to draw this conclusion. Comparison with CxxC motif-deficient mutants of each domain would clarify their role (scaffold or Cu delivery).

As suggested by the reviewer, we have toned down the argumentation on MBD⁻³. Our findings on MBD⁻³ are indeed somewhat speculative. Yes, the MBD⁻² interaction with XtATP7B is most likely weak. It was poorly resolved in the cryo-EM density, and mutation of the proposed interaction network in-between MBD⁻² and the ATPase core had little effect on the XtATP7B function. Similarly, MBD⁻³ is also dynamic in our MD simulations, but the binding position partially overlaps with that of MBD⁻² in XtATP7B. We argue in the manuscript that the fact that a similar position at the A-/P-domain interface is identified (through the XtATP7B structures and in our MD simulations) validates the sites as a possible binding position. However, we also argue in the manuscript that autoinhibition by a MBD bound to this position is more likely in the E1 rather than in the E2P state, as turn-over by the soluble domains of the ATPase core is prevented only in the E1 state (see Fig. 3). Since our core findings on the MBDs concern MBD⁻¹ and MBD⁻² we disagree that additional data are required for further validation of our mechanistic models; it is clearly indicated now that MBD⁻³ may be autoinhibitory.

The description of the atomistic detail of the Cu binding site is interesting and deserves to be described in more detail. There are a few concerns as follows; Although the EM density is shown in the supplements, it is not clear whether the side chains that directly or indirectly contribute to the Cu binding site, including cysteine residues, are resolved enough to identify their rotamers. The authors should provide a better and clearer expanded figure of the density that supports these models.

We have now included a new supplementary figure showing the experimental support for the model of the Cu-binding region (Supplementary figure 22), and also employed a more detailed close-view in Fig. 4.

L444 It seems difficult to discuss detailed distances between side chain sulfur and Cu, as the EM density is not separated. Besides EM density, is there any analysis to determine if these distances are reasonable? Wouldn't a comparison with the coordination of other Cu-binding proteins be useful?

We agree that it is difficult to discuss details regarding distances between sidechains and copper at the determined resolution, and this is why this caveat was and still is clearly mentioned in the old and new versions of the manuscript. We have compared the binding to the one present in the homologous protein AfCopA (also determined in a similar early E1 state), the structure determination of which benefitted from anomalous diffraction signal of heavy metals in the binding region.

L455 The description in this paragraph is important to understand conformational changes of the binding site upon Cu binding, but unfortunately it is difficult to follow this,

especially the relative special orientations between CPC at M4, and Y912 and N913 at M5, only from the information in Fig. 4.

We hope the revised Fig. 4 will assist the reader to understand. We agree that it was difficult to understand this from the previous version of the figure.

Minor comments

Fig. 1. Describe how the two models were superimposed. The difference in Fig 1D is particularly larger than the differences in the other figures, which is confusing for the readers. Comparison of E1 and E2P states from different viewpoints may be helpful to follow its conformational change. Also, in Fig. 1D, A domain in the E2P state is difficult to recognize. For example, the use of light colors corresponding to each domain will help the readers.

The structures in Fig. 1 are aligned using overall alignment. We agree that overall alignment is not optimal for visualizing conformational changes of P-type ATPases, but they do represent easy-to-understand alignments. Fig. 1D has been adjusted according to the suggestion of the reviewer.

L211 Cite Supp Fig 13. It is unclear from this figure that the different position of M3 changes the exposure of the MB platform. Would it be possible to effectively show the different degrees of opening for this area by showing the molecular surface? Also, here the information on how to superimpose these structures is missing, readers (at least for this reviewer) who look at Fig. 1D expect much larger conformational changes in the E1-E2P transition.

We have removed this discussion to increase clarity of the manuscript.

L231-232 As far as mentioning the AF2 model, it should be shown and compared to the model in the figures.

We have introduced a supplementary figure with all mentioned AlphaFold models with a comparison of our structure (supplementary figure 16).

L276 The difference in structure does not appear to be enhanced in LpCopA. Rather, the MB of LpCopA is closer to the ATPase core. Structural comparison in Fig. 2E does not seem to be an appropriate example of what the authors want to describe here.

We disagree with the reviewer. In comparison to the proteins without (classical) MBD (truncated AfCopA in the E1 state, and LpCopA in the E2P conformation, respectively) the MB' platform is closer to the entry site in the E1 state, but further away from the entry site in the E2P configuration (see Fig. 2d,e). Thus, the linker in-between MA and MBD⁻¹ likely enhances the remodeling that occurs with the conformational changes from E1 to E2P.

L310 According to Figure 1CD of reference 22, it seems that MBD-2 (MBD5 in the ref) is better resolved than MBD-1 (MBD6). It is not acceptable to say that the interaction is weak only in this area based on the low resolution of the domain.

From the cryo-EM densities for XtATP7B there is no doubt that the MBD⁻² is significantly worse resolved than most of the ATPase core (the data is publicly available and can easily be assessed). This is also true for MBD⁻¹.

From a structural biological point of view, if one domain/part is poorly resolved while others are not, one of the most common interpretations of such scenarios is that the domain is flexible, which in turn means it is weakly bound. Moreover, as we argued above and in the manuscript, mutation of the proposed interaction network in-between

MBD⁻² and the ATPase core had little effect on the XtATP7B function. Conversely, the equivalent interaction network in-between MBD⁻² and the ATPase core is a known region for disease causing mutations in hATP7B. Note, we do not suggest binding at the A-/P-domain interface does not appear, simply that the functional role of this binding likely is insignificant.

L323 Show the AlphaFold2 model itself in the figure.

All AlphaFold models are now shown in the new supplementary figure 16.

L353 It is unclear what "most of latter" indicates.

This has been rephrased.

L354 Show 8IOY density map as shown in 7XUM in Supplementary Figure 16

The density map for PDB-ID 8IOY is included in supplementary figure 17 (previously supplementary figure 16).

L382 MD Data is not shown in the paper.

This is incorrect. The MD data was and is shown in Fig. 3 as well as in supplementary figure 19 (previously supplementary figure 19).

L386 There is no MBD-3 in Fig 3d

This is correct. That is because MBD⁻³ is flexible in those simulations.

L405 Does Cu-binding to the MBD-3 release the autoinhibition? Is there any evidence or references indicating this? Supplementary Figure 17 only shows that the ATOX1 does not interfere with the ATPase core when previously determined ATOX1-MBD complex structure is superimposed to the assumed structure obtained by the MD run, therefore is not a direct basis for concluding that this causes a release of autoinhibition by MBD-3.

As indicated previously and as is now also more clearly underscored in the manuscript our data on MBD⁻³ (from MD simulations only) are speculative. What we show in the supplementary figure is that ATOX1 can associate with MBD⁻³, also when positioned at the A-/P-domain interface as we infer from our MD simulations. According to the model, ATOX1 and MBD interact CXXC to CXXC and hence copper can be transferred this way. As deduced by for example Bitter et al, 2022, Sci Adv, MBD⁻⁶ to MBD⁻³ are autoinhibitory. Thus, it is attractive to propose that autoinhibition achieved via metal-sensing MBDs is released via copper-donation to N-terminal MBDs (MBD⁻³ in the case of HMA4) from chaperones such as ATOX1. The reviewer is correct that we cannot conclusively conclude this is the case based on the available data, and hence we have changed the language accordingly.

Fig. 3. The Order of displayed panels is confusing, especially for panel F.

The order of the panels follows that of the order in which they are referred to in the text. The panels have been placed in a different order in the figure now.

L509 Why being E1 without Cu would be supported by the AF2 model?

As far as we know all AlphaFold models of P_{1B}-ATPases reside in an E1 like state, which we hence assume represents a ground state.

Figure 3. Please indicate CxxC motif in each MBD domain by sticks or spheres.

This suggestion has been implemented.

Reviewer #1 (Remarks to the Author):

In this revised manuscript, the authors substantively addressed the previous critique. The manuscript is clearer, more informative and much better describes the new structural and mechanistic details which help to resolve several controversial issues. The important structural role of MBD-1 (new information) is well substantiated and the mechanism of copper entry and events within the translocation pathway are much more clear. The four structures and the new mechanistic findings have implications for the entire P1B family of ATP-driven transporters. Overall, the study provides an important contribution to understanding of transmembrane copper transport.

Reviewer #2 (Remarks to the Author):

Now authors addressed all the concerns raised by this reviewer and appropriately revised the manuscript.